# A novel single alpha-helix DNA-binding domain in CAF-1 promotes gene silencing and DNA damage survival through tetrasome-length DNA selectivity and spacer function

Ruben Rosas[1†], Rhiannon R Aguilar[2,3†], Nina Arslanovic[2], Anna Seck[4], Duncan J Smith[4], Jessica K Tyler[2]*, Mair EA Churchill[1,5]*

[1]Program in Structural Biology and Biochemistry, University of Colorado Anschutz Medical Campus, Aurora, United States; [2]Department of Pathology and Laboratory Medicine, Weill Cornell Medicine, New York, United States; [3]Weill Cornell/Rockefeller/Sloan-Kettering Tri-Institutional MD-PhD Program, New York, United States; [4]Department of Biology, New York University, New York, United States; [5]Department of Pharmacology, University of Colorado School of Medicine, Aurora, United States

**\*For correspondence:**
jet2021@med.cornell.edu (JKT);
mair.churchill@cuanschutz.edu
(MEAC)

[†]These authors contributed equally to this work

**Abstract** The histone chaperone chromatin assembly factor 1 (CAF-1) deposits two nascent histone H3/H4 dimers onto newly replicated DNA forming the central core of the nucleosome known as the tetrasome. How CAF-1 ensures there is sufficient space for the assembly of tetrasomes remains unknown. Structural and biophysical characterization of the lysine/glutamic acid/arginine-rich (KER) region of CAF-1 revealed a 128-Å single alpha-helix (SAH) motif with unprecedented DNA-binding properties. Distinct KER sequence features and length of the SAH drive the selectivity of CAF-1 for tetrasome-length DNA and facilitate function in budding yeast. In vivo, the KER cooperates with the DNA-binding winged helix domain in CAF-1 to overcome DNA damage sensitivity and maintain silencing of gene expression. We propose that the KER SAH links functional domains within CAF-1 with structural precision, acting as a DNA-binding spacer element during chromatin assembly.

## Editor's evaluation

The work is an important contribution that advances understanding of CAF-1 function by identifying the KER region as that responsible for DNA size-selective binding of the complex. X-ray crystallography shows the KER region forms an unusual single α-helix (SAH) structure that binds DNA on its own in a size-selective manner. Mutations in the KER helix compromise chromatin assembly and give DNA damage sensitive phenotypes.

## Introduction

In eukaryotes, dynamic local and global chromatin structures regulate accessibility to the genome for all DNA-dependent processes (*Yadav et al., 2018*; *Klemm et al., 2019*). The nucleosome is the fundamental unit of chromatin, comprising two H3/H4 and two H2A/H2B histone dimers wrapped

by approximately 147 bp of DNA (*Luger et al., 1997*). DNA replication requires the disassembly of parental nucleosomes, followed by a highly regulated dynamic assembly process for recycling of parental histones and depositing nascent histones onto the newly replicated DNA (*Smith and Stillman, 1991*; *Escobar et al., 2021*). The histone chaperone chromatin assembly factor 1 (CAF-1) facilitates the dimerization of nascent H3/H4 dimers onto replicated DNA, forming the subnucleosomal structure known as the tetrasome (*Smith and Stillman, 1989*; *Kaufman et al., 1995*; *Mattiroli et al., 2017*; *Sauer et al., 2017*; *Sauer et al., 2018*).

In multicellular organisms, the essential functions of CAF-1 are required for the maintenance of epigenetic landscapes and gene expression patterns (*Cheloufi et al., 2015*; *Cheloufi and Hochedlinger, 2017*; *Smith and Whitehouse, 2012*; *Ramachandran and Henikoff, 2016*; *Sauer et al., 2018*). How CAF-1 deposits H3/H4 at sites of DNA synthesis remains largely unknown. However, the identification of functional domains within CAF-1 has provided significant insight. Three subunits, Cac1, Cac2, and Cac3, form the CAF-1 complex (*Sauer et al., 2018*; *Smith and Stillman, 1989*). Central to the localization of CAF-1 to replicated DNA are interactions with the replisome through PCNA interacting peptides (PIP box) in the Cac1 subunit (*Shibahara and Stillman, 1999*; *Krawitz et al., 2002*). Additionally, the DNA-binding function of the winged helix domain (WHD) located in Cac1 contributes to the recruitment of CAF-1 to sites of replication (*Sauer et al., 2017*; *Zhang et al., 2016*; *Mattiroli et al., 2017*).

DNA-binding studies of the *Saccharomyces cerevisiae* CAF-1 (yCAF-1) revealed a preference for binding to DNA that is at least 40 bp long (*Sauer et al., 2017*), which is slightly less than the length of DNA needed to form a tetrasome (*Luger et al., 1997*; *Donham et al., 2011*). As the WHD binds to short (10–16 bp) DNA fragments (*Zhang et al., 2016*; *Mattiroli et al., 2017*), it is unlikely to confer this DNA-length dependence to CAF-1 or allow for sufficient spacing for the assembly of tetrasomes in vivo. However, the lysine/glutamic acid/arginine-rich (KER) region of the Cac1 subunit has been implicated as a possible second DNA-binding domain (DBD), as CAF-1 lacking the WHD could still bind to DNA (*Sauer et al., 2017*). Whether the KER cooperates with the WHD or PIP box to recruit CAF-1 to sites of DNA synthesis, contributes to the length-dependent DNA recognition of CAF-1, or has other biological roles remains unclear.

Here, we characterized the KER region of yCAF-1 using biophysical, structural, and functional approaches in vitro and in budding yeast. The crystal structure of the KER and DNA-binding experiments revealed a novel single alpha-helix (SAH) domain with DNA-binding ability that we found drives the selectivity of yCAF-1 for tetrasome-length DNA. Features of the structure and sequence of the KER SAH required for DNA binding were defined. In yeast, the KER structure is important for CAF-1-mediated chromatin assembly through cooperation with the WHD during DNA damage repair and gene silencing.

## Results

### The KER is a major DBD in CAF-1 that cooperates with the WHD in vivo

The previously observed DNA-length preference of yCAF-1 (*Sauer et al., 2017*) cannot be explained by the DNA-binding properties of the WHD (*Zhang et al., 2016*; *Mattiroli et al., 2017*; *Sauer et al., 2017*). To determine whether the KER promotes this DNA-length preference, we first expressed and purified recombinant tri-subunit yCAF-1, and the isolated domains KER (Cac1 residues 136–225, yKER), and WHD (Cac1 residues 457–606, yWHD) (*Figure 1a* and *Figure 1—figure supplements 1 and 2a–c*). Using electrophoretic mobility shift assays (EMSAs) we measured the binding affinity for each protein to DNA fragments of different lengths (*Figure 1b–i* and *Figure 1—figure supplement 3a, b*). To determine the DNA-binding affinities, the EMSA images were quantitated and the resulting binding curves fitted with *Equation 1* (see Methods) (*Figure 1j, k* and *Figure 1—figure supplement 3c*), which gave values for the dissociation constant ($K_D$) and cooperativity (Hill coefficient, $h$) (*Figure 1l*). yCAF-1 and the isolated domains bound to DNA with the appearance of a ladder of bands, especially for the yKER that has the largest number of bands, which are related to the length of the DNA fragment (20, 30, 40, and 80 bp). These results indicate that multiple proteins bind to a single DNA fragment (*Figure 1f–i*) and the Hill coefficients suggest a cooperative mode of DNA binding (*Figure 1l*). For DNA of the same length (*Figure 1l*), the yKER had $K_D$ values between 18 and 50 nM, whereas the yWHD bound three to six times more weakly. yCAF-1 had $K_D$ values between those observed

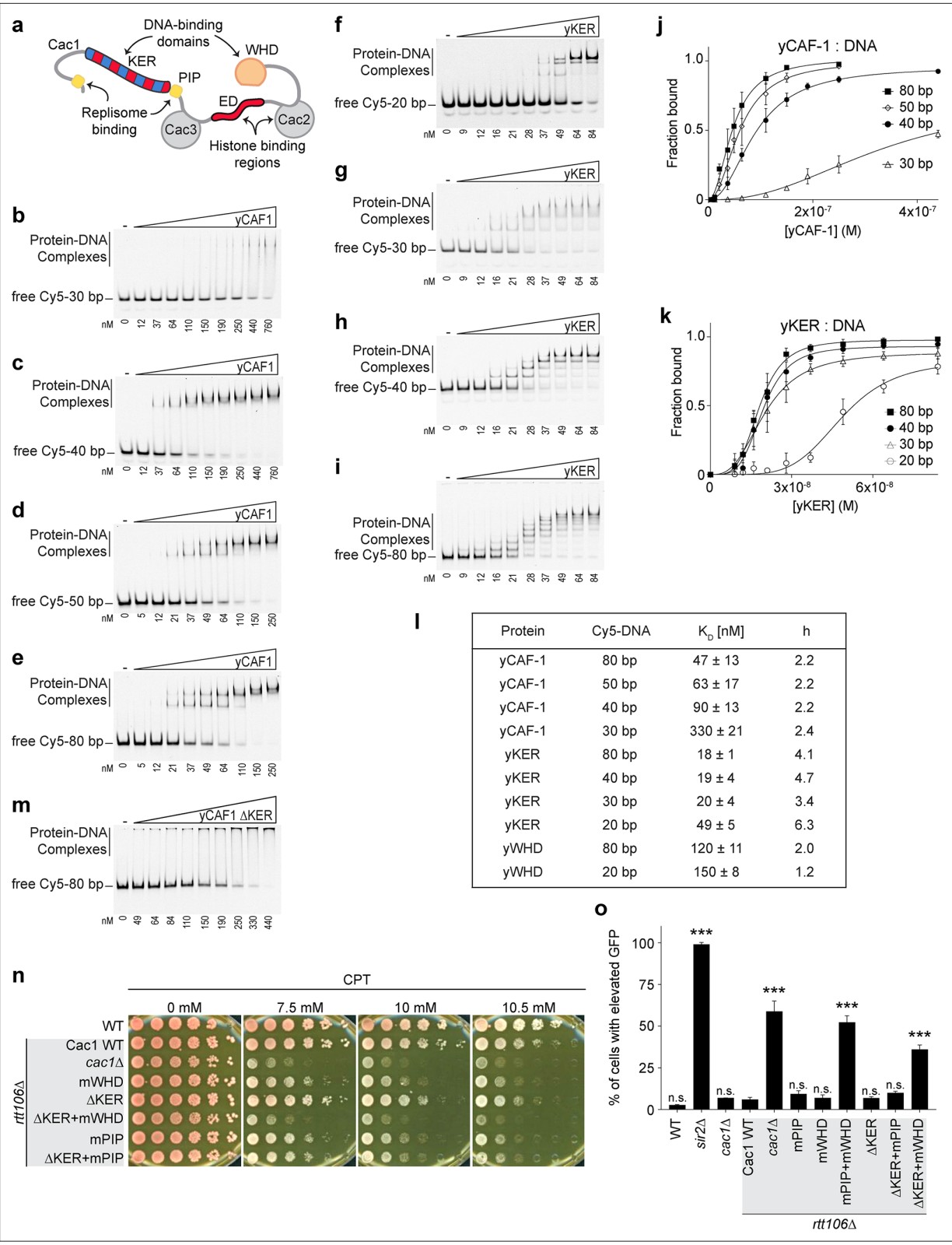

**Figure 1.** The yKER region favors binding to tetrasome-length DNA and facilitates the function of yCAF-1 in vivo. (**a**) Cartoon representing the molecular architecture of the yCAF-1 complex highlighting the protein subunits and functional domains. Domains include the K/E/R-rich (DNA-binding domain), PCNA interacting peptides (PIP boxes), Cac3-binding site (small), E/D-rich regions (histone binding), Cac2-binding site (middle), and a DNA-binding winged helix domain (WHD). (**b–i**) Representative images of electrophoretic mobility shift assay (EMSA) experiments for yCAF-1 or yKER, with

*Figure 1 continued on next page*

*Figure 1 continued*

2 or 3 nM of Cy5-DNA; the range of protein concentrations are: 5–250 and 9–84 nM, respectively. (**j, k**) Quantitative analyses of all EMSAs of yCAF-1 or yKER with Cy5-labeled DNA. Data from at least three independent experiments were plotted as the mean and standard deviation (error bars). The binding curves were fitted using **Equation 1**. (**l**) Table summarizing the dissociation constant ($K_D$) and Hill coefficient (*h*) values obtained from EMSAs of the indicated proteins and DNA. Values were obtained from fitting plots using **Equation 1**. (**m**) Representative image of an EMSA experiment for yCAF-1 ΔKER with 3 nM of Cy5-80 bp DNA; the range of protein concentrations is 49–440 nM. (**n**) Yeast spot assay with fivefold serial dilutions of cultures of the indicated strains; grown in the presence of Camptothecin (CPT) at the specified concentrations. (**o**) Bar graph indicating the percentage of cells exhibiting elevated Green Fluorescent Protein (GFP) levels from yeast cultures of the indicated strains sorted by flow cytometry. Error bars indicate the standard deviation of the calculated values from three measurements. Statistical significance was calculated by Student's *t*-test where ***$p < 0.001$ relative to Cac1 Wild-type (WT) cells. See also **Figure 1—source data 1** and **Figure 1—source data 2**.

The online version of this article includes the following source data and figure supplement(s) for figure 1:

**Source data 1.** The yKER region favors binding to tetrasome-length DNA and facilitates the function of yCAF-1 in vivo.

**Source data 2.** The yKER region favors binding to tetrasome-length DNA and facilitates the function of yCAF-1 in vivo.

**Figure supplement 1.** Proteins and mutants used in this study.

**Figure supplement 2.** Quality of proteins used in this study.

**Figure supplement 2—source data 1.** Quality of proteins used in this study.

**Figure supplement 3.** Chromatin assembly factor 1 (CAF-1) DNA-binding analysis and in vivo assays.

**Figure supplement 3—source data 1.** Chromatin assembly factor 1 (CAF-1) DNA-binding analysis and in vivo assays.

for the yKER and the yWHD, except for weaker binding to the 30-bp DNA fragment ($K_D$ of 330 nM) (*Figure 1l*). This analysis shows that yCAF-1 has a higher affinity for DNA fragments in the range of 50–80 bp, which are DNA lengths considered to be sufficient for tetrasome formation (*Donham et al., 2011*). In contrast, increasing DNA length only slightly decreased $K_D$ values for the yKER or yWHD (*Figure 1l*).

To determine the importance of the KER to the DNA-binding function of yCAF-1, yCAF-1 lacking only the KER, Cac1 residues 136–225 (yCAF1 ΔKER), was produced (*Figure 1—figure supplement 1* and *Figure 1—figure supplement 2a*) and tested using EMSA. Deletion of the KER region impaired DNA binding of yCAF-1 and also resulted in the failure of the complexes to migrate into the gel, possibly due to aggregation (*Figure 1m*). As the yCAF-1 ΔKER mutant still has the WHD, this residual DNA-binding function might be due to the WHD or other unknown DNA-binding regions within CAF-1. Together, these data show that the KER binds more tightly to DNA than the WHD and the presence of the KER in the context of yCAF-1 is needed for high DNA-binding affinity.

As the KER confers the majority of DNA-binding affinity to yCAF-1 in vitro, we investigated the impact of deleting the KER on CAF-1 function in chromatin assembly after DNA replication and repair (*Smith and Stillman, 1991*; *Ye et al., 2003*; *Smith and Stillman, 1989*; *Gaillard et al., 1996*; *Mello et al., 2002*; *Tyler et al., 1999*). Yeast cells harboring defective CAF-1, including loss of the Cac1 subunit, or loss of its WHD, PIP box, or histone-binding regions (*Figure 1a*), have been shown to be more sensitive to DNA damaging

**Table 1.** Relative sensitivity to Camptothecin (CPT) and Zeocin of yeast cells harboring chromatin assembly factor 1 (CAF-1) mutations in a *rtt106Δ* background.

| | CPT sensitivity | Zeocin sensitivity |
|---|---|---|
| Cac1 WT | + | + |
| *cac1Δ* | +++++ | +++++ |
| ΔKER | ++ | ++ |
| mWHD | +++ | +++ |
| mPIP | +++ | ++ |
| mWHD+mPIP | +++++ | +++++ |
| ΔKER+mWHD | +++++ | +++++ |
| ΔKER+mPIP | +++ | ++ |
| Δmiddle-A | ++ | ++ |
| Δmiddle-A+mWHD | +++++ | +++++ |
| 2xKER | ++ | +++ |
| 2xKER+mWHD | ++ | +++ |
| Δ145–149 | + | + |
| Δ145–149+mWHD | ++++ | +++++ |
| Δ225–226 | ++ | ++ |
| Δ225–226+mWHD | +++++ | +++++ |
| Δ225–226+mPIP | +++ | +++ |
| KER::hKER | + | ++ |
| KER::hKER+mWHD | ++++ | +++++ |

agents and have impaired establishment of chromatin landscapes (*Li et al., 2008*; *Zhang et al., 2016*). We examined cell growth and survival of budding yeast mutants in a spot assay in the presence of the DNA topoisomerase I inhibitor Camptothecin (CPT) (*Eng et al., 1988*) and Zeocin. While CPT stabilizes covalently bound DNA topoisomerase I complexes on chromatin resulting in replication fork collisions and DNA double-strand breaks (DSBs) (*Pommier, 2006*), Zeocin intercalates into DNA to induce DSBs independent of the replication process (*Chankova et al., 2007*). yCAF-1 mutants were generated by site-directed mutagenesis of the gene that encodes the Cac1 subunit (*CAC1*) via CRISPR–Cas9 at the endogenous genomic locus in yeast strains deleted for the *RTT106* gene (*rtt106Δ*) that encodes for a histone chaperone with overlapping roles of CAF-1 in yeast (*Huang et al., 2005*; *Li et al., 2008*; *Figure 1—figure supplement 1* and *Figure 1—figure supplement 2d*). Deletion of Cac1 residues 136–225 that encode for the KER region (ΔKER) resulted in a mild sensitivity to CPT and Zeocin (*Figure 1n* and *Figure 1—figure supplement 3d*). Inhibition of the DNA-binding function of the WHD through point mutations in Cac1 residues K564E/K568E (mWHD) in yeast (*Zhang et al., 2016*) showed higher sensitivity to CPT and Zeocin than the deletion of the KER. Strikingly, the double mutant ΔKER+mWHD showed higher CPT and Zeocin sensitivity than either mutant alone, and to a similar extent observed when the Cac1 subunit is absent (*cac1Δ*) (*Figure 1n*, *Table 1*, and *Figure 1—figure supplement 3d*). These results are consistent with roles for both the KER and WHD in CAF-1 function because at least one DBD appears to be sufficient to maintain some CAF-1 function in vivo and overcome repercussions of DSBs, both dependent and independent of DNA replication.

We also investigated a potential coordinated role between the KER and the PIP box for the recruitment of CAF-1 to the replisome. Inhibition of the PIP box downstream of the KER (*Figure 1a*) by substitution of Cac1 residues F233A/F234A (mPIP) (*Zhang et al., 2016*) did not result in an increase in sensitivity to CPT and Zeocin when in combination with ΔKER (ΔKER+mPIP) (*Figure 1n*, *Table 1*, *Figure 1—figure supplement 1*, and *Figure 1—figure supplement 3e*). This is in contrast to mWHD+mPIP cells (*Table 1* and *Figure 1—figure supplement 3e*) where WHD and the PIP box cooperate in their recruitment function (*Zhang et al., 2016*). These results suggest a role of the KER independent of recruitment of CAF-1 to the replisome via the PIP box.

CAF-1 function can influence cellular gene expression profiles, presumably by the deposition of nascent nucleosomes that promote the reestablishment of chromatin landscapes post-DNA replication (*Ramachandran and Henikoff, 2016*). To investigate the role of the KER in this process, we used strains with a Green Fluorescent Protein (GFP) reporter in the *HMR* mating-type locus of budding yeast. In normal conditions the *HMR* locus is silenced but defects in the chromatin result in expression of GFP with measurable fluorescence by flow cytometry (*Huang et al., 2005*; *Laney and Hochstrasser, 2003*; *Figure 1o* and *Figure 1—figure supplement 3f*).

Functionality of this reporter assay was confirmed by deletion of *SIR2* (*sir2Δ*), a subunit of the silent information regulator (SIR) complex required for the establishment of silencing of the *HMR* locus (*Rine and Herskowitz, 1987*). Almost 100% of *sir2Δ* cells exhibited GFP fluorescence (*Figure 1o*). Consistent with the CPT and Zeocin sensitivity assays, we found that ΔKER, mWHD, and mPIP cells had low expression of GFP. As previously reported, a high percentage of *cac1Δrtt106Δ* cells have increased GFP expression and mPIP+mWHD cells have a comparable level of GFP expression, suggesting a near-total loss of CAF-1 function (*Zhang et al., 2016*). No increase in GFP expression is seen in response to combined ΔKER+mPIP cells when compared to mPIP alone (*Figure 1o*). In contrast, ΔKER+mWHD cells had GFP expression, similar to *cac1Δrtt106Δ* cells, suggesting a near-complete loss of CAF-1 function. Finally, to assess the impact of deleting the KER more directly on nucleosome assembly in vivo, we examined histone deposition onto Okazaki fragments during DNA replication as we have shown previously that the length of Okazaki fragment lengths is determined by histone deposition into nucleosomes and is disrupted upon deletion of *CAC1* (*Smith and Whitehouse, 2012*). We compared CAF-1 mutants in the WT yeast background and in yeast lacking Rtt106. We found that the Okazaki fragment length distributions of the ΔKER mutant were indistinguishable from that of WT while that of *cac1Δ* was disrupted (*Figure 1—figure supplements 1 and 3g*). That we did not detect effects on Okazaki fragment lengths for the yCAF-1 mutants lacking the intact KER is consistent with the results of the viability and silencing assays for KER mutants, which also retained the WHD. Strikingly, the Okazaki fragments from *rtt106Δ cac1Δ* yeast were highly disrupted (*Figure 1—figure supplements 1 and 3g*) further highlighting the redundancy between Rtt106 and Cac1 for assembling histones onto newly replicated DNA. Together, these results suggest that yCAF-1 requires the

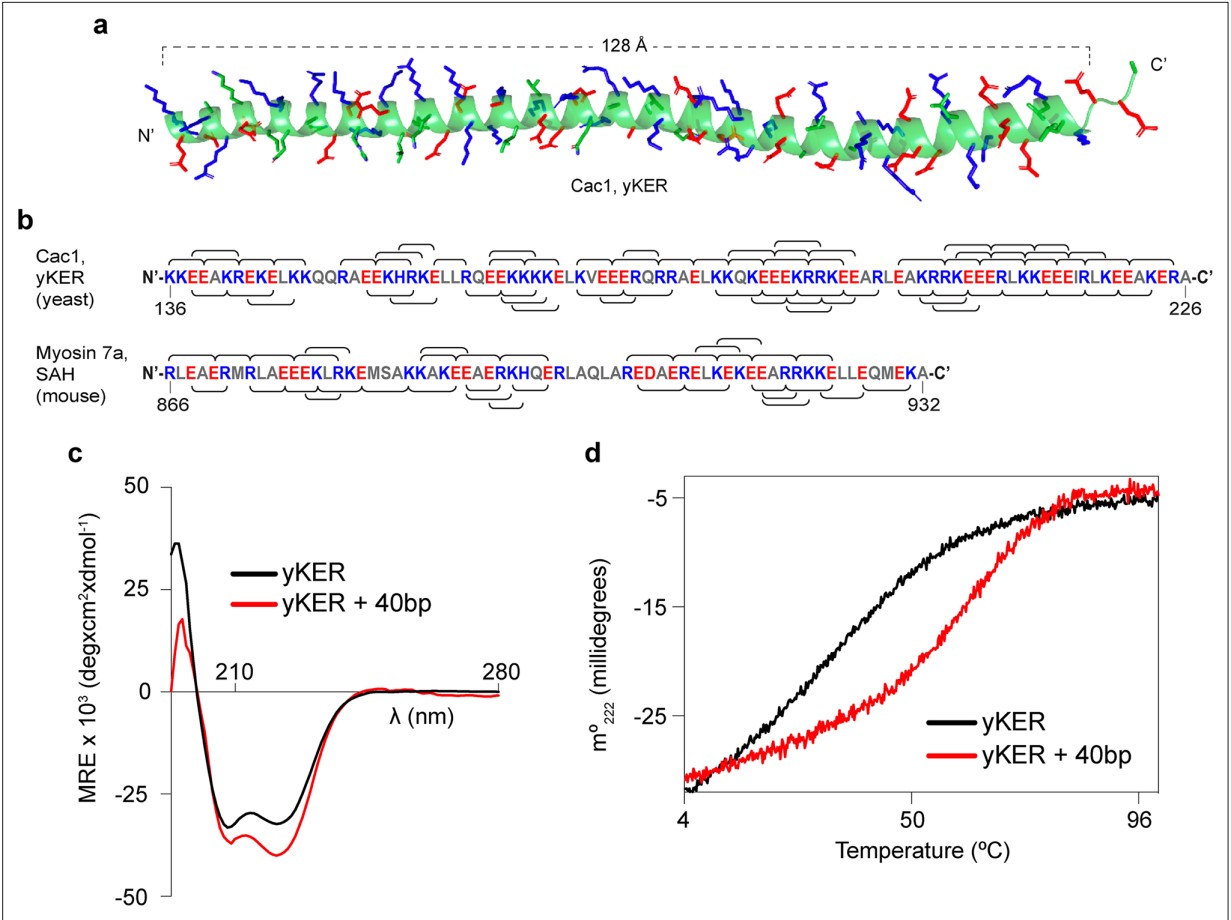

**Figure 2.** The yKER is a single alpha-helix (SAH) domain that forms a stable complex with DNA. (**a**) Ribbon representation of the X-ray crystal structure of yCAF-1 KER region. Cac1 residues 136–222 are shown with side chains of residues Lys, Arg, and His colored in blue and Glu in red. (**b**) Schematic diagram of the indicated SAH sequences with positively charged residues Arg, Lys, and His, colored in blue; and negatively charged residue Glu and Asp colored in red. The brackets along the sequence represent predicted interhelical i, i+4 or i, i+3 ion pairs. (**c**) Overlap of circular dichroism spectra of yKER alone and in the presence of 40 bp DNA. DNA signal was subtracted from the yKER + 40 bp DNA sample to observe only changes in the protein component. (**d**) Thermal denaturation monitored by circular dichroism at 222 nm ($m°_{222}$) of yKER alone and in the presence of 40 bp DNA. See also *Figure 2—source data 1*.

The online version of this article includes the following source data and figure supplement(s) for figure 2:

**Source data 1.** The yKER is a single alpha-helix (SAH) domain that forms a stable complex with DNA.

**Figure supplement 1.** Analysis of the maltose-binding protein (MBP)-yKER crystal structure.

**Figure supplement 1—source data 1.** Maltose-binding protein (MBP)-yKER oligomeric properties.

**Figure supplement 2.** DNA-binding properties of the yKER.

**Figure supplement 2—source data 1.** DNA-binding properties of the yKER.

coordinated action of the KER and WHD to prevent DNA damage sensitivity and maintain silencing of chromatin in vivo. Since both the KER and the WHD bind to DNA, they might have overlapping yet complementary functions in targeting or aligning CAF-1 correctly to the DNA during nucleosome assembly.

## The KER is a first-in-class SAH DBD

To gain insight into how the KER region binds to DNA and facilitates CAF-1 function in vivo, we determined the structure of the yKER at 2.81 Å resolution. Crystals grown of a maltose-binding protein (MBP)–KER fusion construct (*Figure 2a, Figure 2—figure supplement 1a, b, and Supplementary file 4*) contained four copies of the MBP-yKER per asymmetric unit. The yKER extended from the MBP as an almost entirely solvent exposed continuous alpha-helix (*Figure 2—figure supplement 1b*). The

longest yKER helix modeled (PDB 8DEI; molecule A) is 128 Å long with 23 helical turns, encompassing Cac1 amino acids 136–221 (*Figure 2a*). Superposition of the four KER copies gives root mean square deviation (RMSD) values between 0.66 and 2.2 Å, as the helices have a marked curvature in the region of aa 165–190 and larger deviations at the termini (*Figure 2—figure supplement 1c*). In the structure, we saw no evidence of interactions between yKER helices or formation of any tertiary structure, consistent with the definition of an SAH domain (*Wolny et al., 2017*). Inspection of the sequence of the yKER reveals a pattern of opposite-charged residues, lysine/arginine and glutamic acid, that are three or four residues apart (*Figure 2b*), capable of forming an ion pair network. Such a network confers stability to the helix so that SAH domains can be completely solvent exposed in solution (*Batchelor et al., 2019*; *Sivaramakrishnan et al., 2008*; *Wolny et al., 2017*; *Swanson and Sivaramakrishnan, 2014*); such as in the well-characterized Myosin 7a SAH (*Barnes et al., 2019*). The circular dichroism (CD) spectrum of the KER showed characteristics of only alpha helical secondary structure, including the positive absorption band at 195 nm and two negative bands at 208 and 222 nm (*Figure 2c*), validating in solution the structure of the KER observed in the crystal. Interestingly, binding to DNA increased the alpha helical content of the yKER (*Figure 2c*) without changes in the CD signal from the DNA (*Figure 2—figure supplement 2a*), suggesting that no major DNA structural changes occur.

Fundamental to the definition of SAH domains is that they exhibit non-cooperative denaturation transitions due to a lack of tertiary structure (*Wolny et al., 2017*; *Wolny et al., 2014*; *Süveges et al., 2009*). This behavior is also observed in the thermal denaturation monitored by CD at 222 nm of the yKER (*Figure 2d*). Furthermore, chemical crosslinking with disuccinimidyl suberate (DSS) showed no evidence of KER multimers (*Figure 2—figure supplement 1d*), supporting the conclusion that the yKER does not form a tertiary structure. In contrast, in the presence of 40 bp DNA, the thermal denaturation of the yKER showed an increase of 10°C in the melting temperature and a two-state cooperative unfolding transition (*Figure 2d*), which demonstrates the ability of the yKER to form a stably folded protein–DNA complex. The monomeric state of the yKER also suggests that the additional yKER–DNA complexes observed in the EMSAs (*Figure 1f–i*) are due to the addition of yKER monomers to the same molecule of DNA. This was substantiated through use of a heterogenous subunit EMSA (*Hope and Struhl, 1987*; *Gangelhoff et al., 2009*) with the yKER, MBP-yKER, and a combination of the two proteins. The mixed-protein subunit:DNA complexes (*Figure 2—figure supplement 1e*) can only be explained if the KER forms no obligate oligomers on the DNA. Rather, multiple monomers of yKER are recruited to the same molecule of DNA, creating the multiplicity of bands in the EMSA. Finally, DNA binding has not been reported for other SAH domains even though they have a similar amino acid composition (*Figure 2b* and *Figure 2—figure supplement 2c, d*). Examination of a purified Myosin 7a SAH (*Barnes et al., 2019*; *Figure 1—figure supplement 2b* and *Figure 2—figure supplement 2b*) using EMSA detected no DNA binding, indicating that the DNA-binding properties of the yKER are not a general feature of SAH domains.

## The yKER requires both the alpha helical structure and positively charged residues for DNA binding and yCAF-1 function in vivo

As the yKER is a novel DNA-binding SAH, it was important to define which sequence features are responsible for the DNA-binding functionality. We noticed that the yKER SAH is particularly enriched in positively charged residues, unlike the Myosin 7a SAH (*Figure 2—figure supplement 2c, d*). These residues are biased toward one face of the yKER helix and confer a net positive charge along most of the length of the yKER, which is concentrated toward the N-terminal and middle regions (*Figure 3a*). To map the DNA-binding region, we designed and purified five truncated versions of the yKER (*Figure 3a* and *Figure 1—figure supplement 2b*). These proteins exhibited different alpha helical content, which was markedly greater for the constructs containing C-terminal regions (*Figure 3b*). However, only the middle-A protein (residues 155–204) exhibited strong DNA binding similar to the intact KER (*Figure 3a, c–g*), whereas the N-half and middle-B constructs, which partially overlap with middle-A, did not bind well to DNA (*Figure 3e, g*). Although regions of the yKER outside of the middle-A region likely contribute to KER function, the middle-A region is required for DNA binding due to both the positively charged residues and alpha helical structure. Notably, deletion of the middle-A residues from yCAF-1 (yCAF1 Δmiddle-A) abrogates binding of yCAF-1 to DNA (*Figure 3h* and *Figure 3—figure supplement 1*), confirming the importance of this region. Interestingly, we also observed that yCAF1 Δmiddle-A binds to DNA worse than the deletion of the KER (yCAF1 ΔKER)

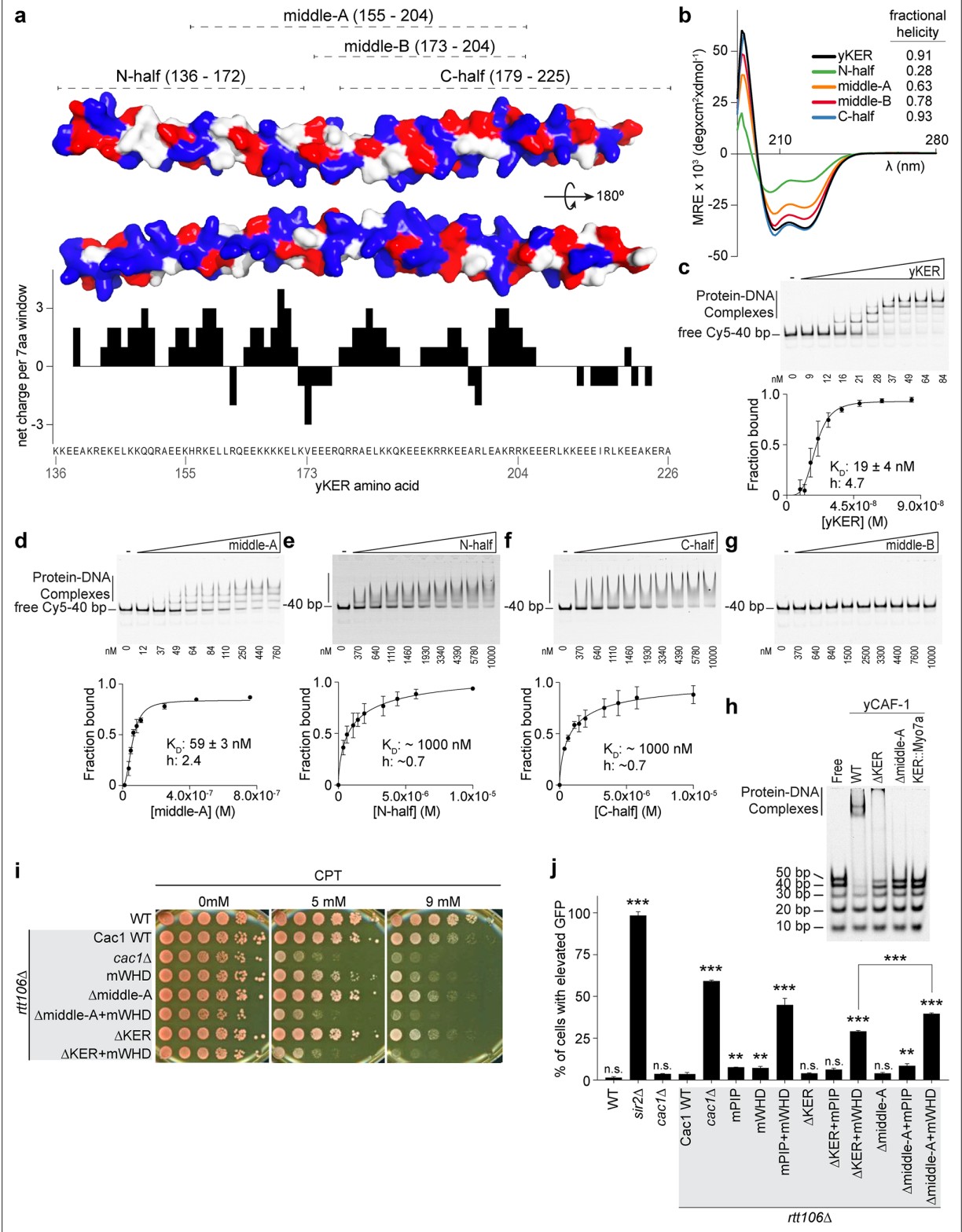

**Figure 3.** The yKER middle region is required for DNA binding and yCAF-1 function in vivo. (**a**) Surface representation of two views of the yCAF-1 KER structure with basic residues colored in blue, acidic in red, and polar or hydrophobic in gray. The dashed lines at the top illustrate the yKER truncations under investigation. The bar graph represents the net charge calculated for a sliding window of seven amino acids along the yKER sequence. The resulting net charge was assigned to the fourth residue in the window. (**b**) Overlay of circular dichroism spectra of the yKER constructs indicated in (**a**). (**c–g**) Representative images of electrophoretic mobility shift assay (EMSA) experiments and binding curves for the yKER constructs indicated in (**a**)

*Figure 3 continued on next page*

*Figure 3 continued*

Cy5-40 bp DNA (either 2 or 2.5 nM) binding was observed over a range of protein concentrations of 9–84 nM for the yKER, 12–760 nM for the middle-A, 0.37–1 μM for N-half, C-half, and middle-B. $K_D$ and $h$ values were calculated from binding curves fitted with *Equation 1* and were plotted as the mean of at least three independent experiments. (**h**) Representative image of an EMSA showing the binding of a fixed concentration (250 nM) of yCAF-1, yCAF-1 ΔKER, yCAF-1 Δmiddle-A, and yCAF-1 KER::Myo7aSAH proteins binding to a set of different length of Cy5-labeled DNA fragments at 1 nM each. (**i**) Yeast spot assay with fivefold serial dilutions of cultures of the indicated strains; grown in the presence of Camptothecin (CPT) at the specified concentrations. (**j**) Bar graph indicating the percentage of cells exhibiting elevated Green Fluorescent Protein (GFP) levels from yeast cultures of the indicated strains sorted by flow cytometry. Error bars indicate the standard deviation of the calculated values from three measurements. Statistical significance was calculated by Student's *t*-test where **p < 0.01 and ***p < 0.001 are relative to Cac1 WT cells. See also *Figure 3—source data 1*.

The online version of this article includes the following source data and figure supplement(s) for figure 3:

**Source data 1.** The yKER middle region is required for DNA binding and yCAF-1 function in vivo.

**Figure supplement 1.** The yKER middle region is required for DNA binding.

(*Figure 3h*), suggesting that residual SAH structure might not support the correct organization or architecture of yCAF-1 in a manner that hinders the activity of the WHD. In the context of CAF-1, a substitution of the KER for the Myosin 7a SAH (KER::Myo7a) in yCAF-1, also abolished DNA binding (*Figure 3h*). In vivo, Δmiddle-A yeast behave like the ΔKER cells, exhibiting similar sensitivity to CPT and Zeocin, and even higher GFP expression levels in the silencing assay when in combination with mWHD (*Figure 3i, j, Table 1*, and *Figure 2—figure supplement 2e*). Taken together, we conclude that a specific region within the KER domain (middle-A) simultaneously forms an alpha-helix and engages with DNA to drive DNA binding and the biological functions of yCAF-1.

## The KER confers the selectivity of CAF-1 for tetrasome-length DNA

Consistent with previous results (*Sauer et al., 2017*), and in *Figure 1*, yCAF-1 was found to have the highest binding affinity for DNA fragments that are at least 40 bp in length, which suggests a DNA-length selective property of CAF-1 that is driven by the KER. To test this hypothesis, we developed a novel EMSA approach, which uses equimolar concentrations of Cy5-labeled DNA fragments spanning 10–50 bp in length (Cy5-DNA ladder) to detect the DNA-length preferences of CAF-1 under competition conditions between different lengths of DNA (*Figure 4a–e*). The free DNA signal for each fragment in the Cy5-DNA ladder was quantitated and plotted as a function of protein concentration. Subjecting yCAF-1 to this assay resulted in a DNA-length selective binding behavior, as shown by the preferential depletion of the 50 and 40 bp DNA fragments from the free DNA at lower concentrations of yCAF-1 compared to the 20 and 30 bp fragments (*Figure 4a*), while 10 bp remained unchanged. In order to visualize these differences in competition, we plotted the apparent $K_D$ ($K_{Dapp}$) as a means to compare different proteins and different DNA lengths in the assay (*Figure 4f*). We observed a threshold effect whereby CAF-1 binds to DNA of 40 bp in length length or longer with a similar $K_{Dapp}$, but there were mutants that had increased binding affinity for 50 bp compared to 40 bp. This can be seen by plotting the rate of the apparent dissociation constant change from 40 to 50 bp, which allows for these thresholds to be easily distinguished (*Figure 4f, g*). Similarly, the yKER bound to the Cy5-DNA ladder in a DNA-length-dependent manner. However, the yKER substantially depleted 30 and 20 bp fragments from the free DNA at lower concentrations (*Figure 4b*), revealing a very slight threshold effect compared to yCAF1. In contrast, the yWHD exhibited virtually no DNA-length selective binding, except that like the KER, the shortest 10 bp DNA was bound poorly (*Figure 4c*). yCAF-1 with the WHD deleted (Cac1 amino acids 520–606) (yCAF1 ΔWHD) did not bind well to the 10–30 bp fragments, but did have a slight preference for the 50 bp over the 40 bp DNA (compare *Figure 4a, d*). Relevant to this observation, previous studies revealed that in a truncated form of CAF-1, the WHD interacts with the ED region (*Figure 1a*) and that H3/H4 dimer binding to the ED was needed to displace the WHD and free it to bind to DNA (*Mattiroli et al., 2017*). This 'autoinhibited state' of the WHD can also be released through substitution of Cac1 residues 397–431 in the ED region by a glycine/serine/leucine linker (ED::GSL), which decreases the affinity of the WHD for the ED and allows the WHD to bind DNA in a histone-free manner (*Mattiroli et al., 2017*). Therefore, we recapitulated the ED::GSL substitution in the context of full-length yCAF-1 (yCAF1 ED::GSL) (*Figure 4e*) and subjected it to the Cy5-DNA ladder assay. We found that yCAF1 ED::GSL showed a slight preference for the 50 bp DNA compared to the 40 bp (*Figure 4e–g*), suggesting that the uninhibited WHD contributes further to the length of DNA recognized by CAF-1. Together, these results demonstrate

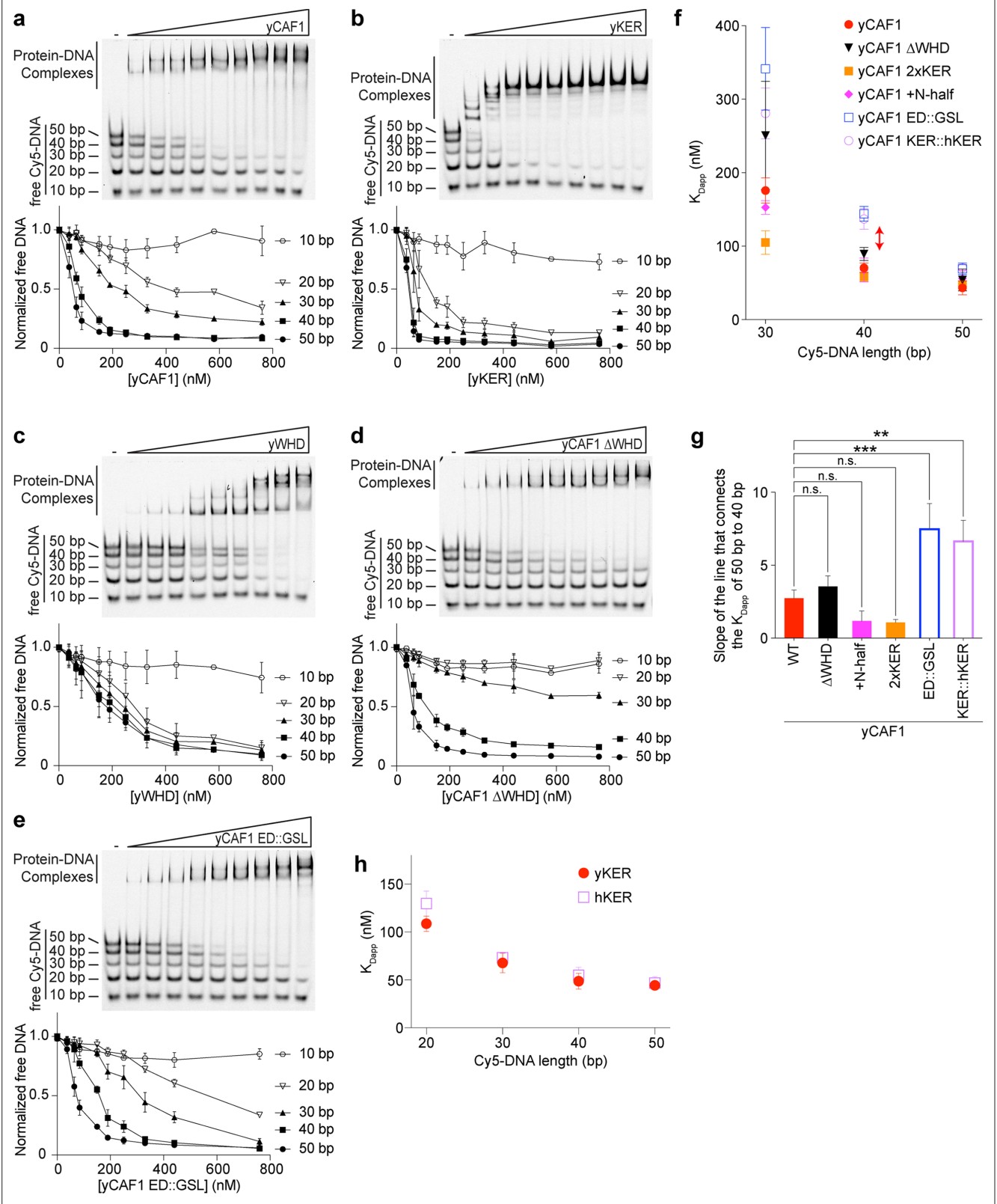

**Figure 4.** The yKER confers DNA-length selectivity to yCAF-1. (**a–e**) Representative images of electrophoretic mobility shift assays (EMSAs) showing DNA binding of yCAF-1, yKER, yWHD, yCAF-1 ΔWHD, or yCAF-1 ED::GSL where each Cy5-labeled DNA fragment is at 1 nM concentration and the range of protein concentration was 37–760 nM for all constructs. Below each gel image, the graph shows the quantitation of free (unbound) DNA signal for each Cy5-labeled DNA as a function of protein concentration. The data are plotted as the mean and standard deviation from at least three

*Figure 4 continued on next page*

Figure 4 continued

measurements. (f) Plots representing the apparent $K_D$ ($K_{Dapp}$) of the individual DNA fragments from the Cy5-DNA ladder for the indicated yCAF-1 constructs. (g) Plot of the rate (slope) of change of the apparent dissociation constant from 40 to 50 bp. One-way ANOVA analyses show significant differences for the ED:GSL and KER:hKER mutants (**p < 0.01 and ***p < 0.001). (h) Plot representing the protein concentration required to achieve 50% depletion of the individual DNA fragments from the Cy5-DNA ladder for the indicated chromatin assembly factor 1 (CAF-1) domains. See also *Figure 4—source data 1*.

The online version of this article includes the following source data for figure 4:

**Source data 1.** The yKER confers DNA-length selectivity to yCAF-1.

that the KER, but not the WHD, equips yCAF-1 with a DNA-length selectivity function that favors binding to DNA fragments that are at least 40 bp in length. Also, the presence of the WHD permits yCAF1 binding to 20 and 30 bp DNA, suggesting it has either a direct or indirect contribution to DNA binding of yCAF-1; and that there is a further contribution to the length threshold in the state where the WHD is released from the ED.

## The length of the KER alters CAF-1 DNA-length recognition and modulates yCAF-1 functions in vivo

The difference in DNA-length selectivity between the KER in isolation compared to yCAF-1 (*Figure 1l* and *Figure 4a, b*) suggests that the context of the KER within the CAF-1 complex directs preferential binding to the longer DNA fragments. To investigate this, we made perturbations to the length of the KER in yCAF1 and evaluated the effects on DNA binding and function in vivo.

We expressed and purified a yCAF-1 mutant that contains two KER domains in tandem (yCAF1 2xKER) by introducing an additional yKER (amino acids 136–225) in the Cac1 subunit immediately after the endogenous 225 residue (*Figure 1—figure supplement 2a*). Examination of yCAF1 2xKER in our Cy5-DNA ladder experiment showed that unexpectedly the additional KER did not alter the DNA-length threshold of 40 bp (*Figure 4f, g* and *Figure 5a*). A similar result was observed with a yCAF-1 mutant that contains an additional KER N-terminal half of the KER (Cac1 residues 136–172) added after the endogenous 225 residue of Cac1 (yCAF1 +N-half) (*Figure 4f* and *Figure 5—figure supplement 1a*). In vivo, yCAF1 2xKER (2xKER) exhibited mild sensitivity to CPT (*Figure 5b*) and Zeocin (*Figure 5—figure supplement 1b*) to a similar extent as seen for ΔKER cells. Surprisingly, 2xKER in combination with inhibition of the WHD (2xKER+mWHD) did not have an additive effect unlike ΔKER in both CPT and Zeocin conditions (*Figure 5b* and *Figure 5—figure supplement 1b*). Likewise, the 2xKER mutant caused no significant loss of silencing in yeast and it did not increase this effect when in combination with mWHD (*Figure 5c*). Together, addition of a KER sequence did not result in substantial differences in the DNA-length selectivity of yCAF-1, and did not impact the ability to overcome DNA damage and maintain gene silencing in vivo.

We then explored whether truncations of the KER helix have an effect in CAF-1 function in vivo. We made a short deletion within the basic region of the KER (Δ145–149) (*Figure 3a*), which removes five amino acid residues, shortens the helix by 1.4 turns and changes the phase of the helix. The deletion is not in the main DNA-binding region of the KER (*Figure 3*) and is not expected to alter the DNA-binding function of the KER. Interestingly, Δ145–149 cells behave like ΔKER cells, exhibiting similar sensitivity to CPT and Zeocin, as well as high GFP expression levels in our silencing assay when in combination with mWHD (*Figure 5c, d*, *Table 1*, and *Figure 5—figure supplement 1b*). Strikingly, identical results were found when only the deletion of the last two C terminal residues 225–226 (Δ225–226) of the KER region in the Cac1 subunit were made (*Figure 5e, c*, *Table 1*, and *Figure 5—figure supplement 1b, c*). We conclude that a very specific length and/or orientation of the yKER helix within CAF-1 is critical to overcome DNA damage and maintain gene silencing in vivo.

## The longer human KER alters DNA-length-dependent binding and does not substitute for the yKER in yCAF-1 in vivo

Our results from altering the length of the KER of CAF-1 in yeast revealed that CAF-1 function is highly sensitive to the length or phase of the helix, where the deletion of only two residues was sufficient to impair yCAF-1 function similar to a complete deletion of the KER (*Figure 5*). The length of the KER regions in other species differ. In the human homolog CHAF1A, the KER has a similar distribution of

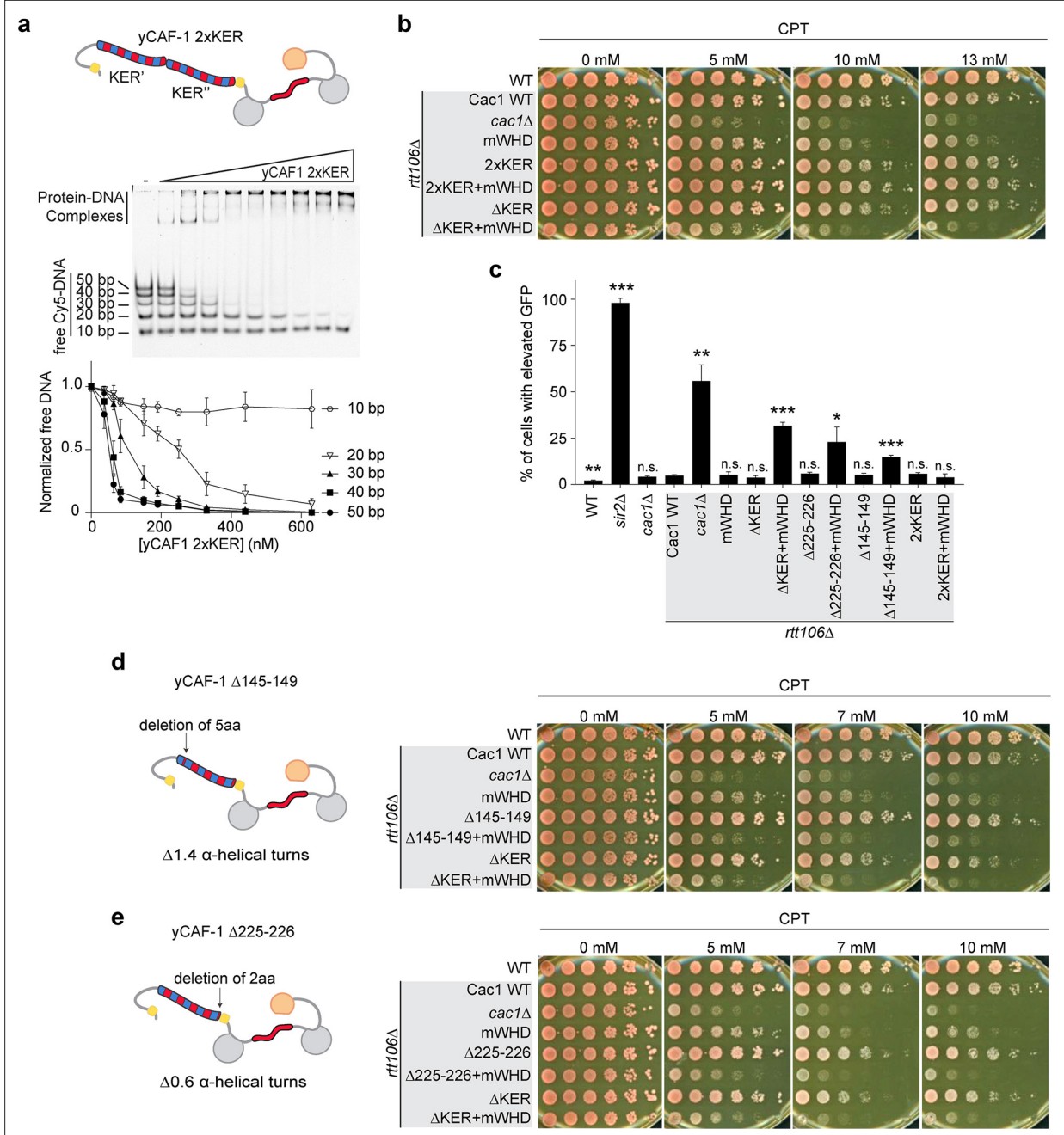

**Figure 5.** The length and the phase of the yKER single alpha-helix (SAH) modulate yCAF-1 functions in vivo. (**a**) Cartoon representing the yCAF-1 2xKER construct along with a representative image of an electrophoretic mobility shift assay (EMSA) showing binding to a set of Cy5-labeled DNA fragments (1 nM each) with a range of protein concentrations from 37 to 630 nM. Below the EMSA image, the free (unbound) DNA signal for each Cy5-labeled DNA is plotted as a function of the protein concentration. The error bars are the standard deviation from at least three measurements. (**b**) Yeast spot assays with fivefold serial dilutions of cultures of the indicated strains; grown in the presence of Camptothecin (CPT) at the specified concentrations. (**c**) Bar graph indicating the percentage of cells exhibiting elevated Green Fluorescent Protein (GFP) levels from yeast cultures of the indicated strains sorted by flow cytometry. Error bars indicate the standard deviation of the calculated values from three measurements. Statistical significance was calculated by Student's t-test where *p < 0.05, **p < 0.01, and ***p < 0.001 relative to Cac1 WT cells. (**d, e**) Yeast spot assays as in (**b**) where cartoons on the left represent the shift of alpha helical turns for the indicated KER deletions in yCAF-1. See also **Figure 5—source data 1**.

The online version of this article includes the following source data and figure supplement(s) for figure 5:

**Source data 1.** The length and the phase of the yKER single alpha-helix (SAH) modulate yCAF-1 functions in vivo.

**Figure supplement 1.** Analysis of the yKER length.

**Figure supplement 1—source data 1.** Analysis of the yKER length.

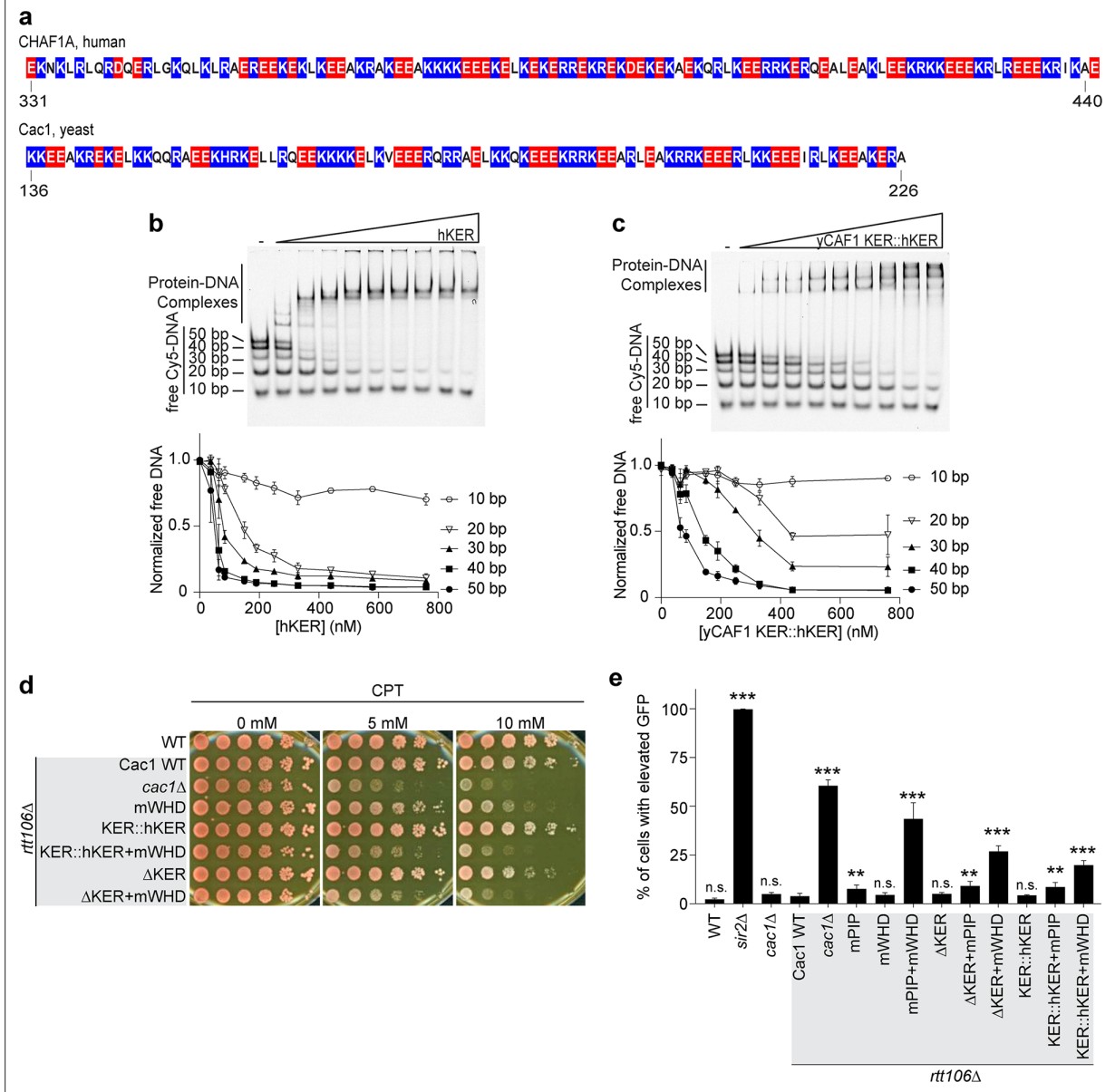

**Figure 6.** Chromatin assembly factor 1 (CAF-1) DNA-length selectivity by the KER is species specific and its function is not conserved in vivo. (**a**) Sequence of the KER region from human (CHAF1A, top) and yeast (Cac1, bottom) homologs with positively charged residues Arg and Lys colored in blue, and negatively charged residue Glu and Asp colored in red. (**b, c**) Images of representative electrophoretic mobility shift assays (EMSAs) of human KER (hKER) and yCAF-1 KER::hKER were each Cy5-labeled DNA fragment is at 1 nM concentration and the range of protein concentration was 37–760 nM for both constructs. The graphs below show the quantitation of free (unbound) DNA signal for each Cy5-labeled DNA as a function of protein concentration. The data are plotted as the mean and standard deviation (error bars) from at least three measurements. (**d**) Yeast spot assay with fivefold serial dilutions of cultures of the indicated strains; grown in the presence of Camptothecin (CPT) at the specified concentrations. (**e**) Bar graph indicating the percentage of cells exhibiting elevated Green Fluorescent Protein (GFP) levels from yeast cultures of the indicated strains sorted by flow cytometry. Error bars indicate the standard deviation of the calculated values from three measurements. Statistical significance was calculated by Student's *t*-test where **$p < 0.01$ and ***$p < 0.001$ relative to Cac1 WT cells. See also *Figure 6—source data 1*.

The online version of this article includes the following source data and figure supplement(s) for figure 6:

**Source data 1.** DNA-length selectivity by the KER is species specific and its function is not conserved in vivo.

**Figure supplement 1.** Analysis of the substitution of the yKER with the hKER.

**Figure supplement 1—source data 1.** Analysis of the substitution of the yKER with the hKER.

basic residues, but it is at least 20 residues longer than the yKER (*Figure 6a*). To address the hypothesis that the longer hKER might have different DNA-binding properties than the yKER, we expressed and purified the hKER (CHAF1A residues 331–441) and using CD found that it has high alpha helical content, and exhibits non-cooperative unfolding as expected for an SAH (*Figure 1—figure supplement 2b* and *Figure 6—figure supplement 1a–c*). Subjecting the hKER to our Cy5-DNA ladder assay resulted in identical DNA-length selectivity behavior to the yKER (*Figure 6b* compared to *4b*). However, substitution of the yKER for the hKER in yCAF-1(yCAF1 KER::hKER) (*Figure 1—figure supplement 2a*) slightly altered the DNA-length selectivity function of yCAF-1, as seen by the slight difference between the depletion of the 50 bp DNA fragments relative to the shorter DNA in our Cy5-DNA ladder assay (*Figure 6c* compared to *Figure 4a*). This difference is largely due to the decreased competitiveness of the 30 and 40 bp DNA compared to the 50 bp DNA (*Figure 4f, g*). This is only observed in the context of yCAF-1 but not in the isolated KERs (*Figure 4h*) and suggests that the longer hKER can alter the DNA-length preference of yCAF-1. Surprisingly, substitution of the KER for the hKER in yeast had a similar effect as the deletion of the KER, with similar sensitivity to CPT and Zeocin, as well as impaired chromatin silencing when in combination with mWHD (*Figure 6d, e* and *Figure 6—figure supplement 1d*). These results indicate that the hKER cannot substitute the yeast KER in vivo. Collectively, these results confirm the expected conservation of the overall SAH and DNA-binding characteristics of the KER from different species. Furthermore, and in agreement with our results in *Figure 5*, the length of the KER alters the DNA-length selectivity and plays a critical role in CAF-1 biological functions.

## Discussion

The results presented here reveal the structure, activities, and function of the KER domain in CAF-1. The KER is a long SAH motif, with a distinct and unique pattern of basic residues. It binds DNA in a non-sequence-specific manner with a binding affinity in the nM range, establishing it as a first-in-class DBD, namely the 'SAH-DBD'. Moreover, we found that in the context of yCAF-1, the KER is largely responsible for observed DNA-length preference of CAF-1 for tetrasome-length DNA (*Sauer et al., 2017*) and CAF-1 function in vivo.

### The KER SAH is a novel DNA-binding motif

A defining feature of the SAH motif is the sequence pattern, which produces stabilizing electrostatic interaction networks along the alpha-helix. The KER SAH is the longest SAH described to date (Cac1 aa 136–226; 90 aa) with the same general pattern of alternating basic and acidic amino acid residues as classic SAHs (*Figure 2*; *Sivaramakrishnan et al., 2008*; *Batchelor et al., 2019*; *Dudola et al., 2017*; *Gáspári et al., 2012*). However, the KER also has a stripe of basic residues along most of the helix, so far only noticeable in the KER SAH of CAF-1. Also unlike the canonical SAH motif, such as in Myosin 7a, the KER SAH binds to DNA. The appearance of multiple KER–DNA complexes in EMSA supports the role of the KER as a non-sequence-specific DBD (*Churchill et al., 1999*; *Churchill et al., 1995*), consistent with the role of CAF-1 in depositing H3/H4 dimers throughout the genome. The KER of CAF-1 is the first SAH-DBD of its type within the larger group IV of 'other alpha-helix DNA-binding domains' (*Luscombe et al., 2000*).Thus, the discovery that the KER is a DNA-binding SAH expands the repertoire of DBDs.

The molecular mechanism of KER-DNA recognition requires both key positively charged residues and alpha helical conformation in the Cac1 middle-A section (*Figure 3*). Although CAF-1 prefers to bind to tetrasome-length DNA (*Luger et al., 1997*; *Donham et al., 2011*), the KER SAH is capable of binding to DNA independently and cooperatively in a final ratio related to the DNA length with an estimated site size as short as 20 bp, which is typical of many DBDs. Many alpha helical DNA-binding motifs, including leucine zippers and helix–loop–helix motifs, bind across the DNA within the major groove (*Luscombe et al., 2000*; *Wolberger, 2021*; *Churchill and Travers, 1991*). Alternatively, the positively charged face of the KER could align with the DNA, similar to the long helices that lie parallel to the DNA exist in chromatin remodelers, such as the HSA and post-HSA domains in the actin-related proteins (Arp4 and Arp8) of INO80 (*Knoll et al., 2018*; *Baker et al., 2021*). However, these helices simultaneously interact extensively with other polypeptides in addition to the DNA (*Knoll et al., 2018*; *Baker et al., 2021*). In the context of the intact CAF-1 complex, there are fewer

CAF-1–DNA complexes observed, which suggests that KER–DNA binding is constrained by other interactions within the complex. Whether the KER binds along the length of the DNA or engages only short stretches of the DNA in a similar manner to the other helical motifs is not clear. Our results are consistent with aspects of both of these models (*Figure 7—figure supplement 1a, b*), as the middle region of the KER confers the ability to bind to DNA, and the positively charged amino acids along one face of the SAH-DBD would be suitable for electrostatic steering as well as recognition of long segments of DNA.

## Function of the KER SAH-DBD in CAF-1 and implications for nucleosome assembly

The KER confers the preference of CAF-1 for tetrasome-length DNA (*Figures 1 and 4*), in spite of the SAH-DBD recognition of shorter DNA lengths (*Figure 4*). Surprisingly, nearly all perturbations to the KER that we tested in yeast had reduced resistance to DNA damage and loss of gene silencing in combination with the mWHD (*Figures 1, 5, 6* and *Table 1*), even a deletion of two residues, which rotated the helix approximately 200° relative to the DNA or other regions of CAF-1. Although doubling or extending the length of the yKER in yCAF-1 did not change the selectivity for 40 bp DNA, the 2xKER strain overall survived CPT and Zeocin induced DNA damage and maintained gene silencing in the absence of a functional WHD. This suggests that the second KER SAH functionally substitutes for the WHD in vivo. We also increased the DNA-length preference of yCAF-1 to 50 bp by either derepressing the DNA-binding function of the WHD, via the ED:GSL mutation or substituting the yKER with the hKER (*Figures 4 and 6*). Both of these mutations provide additional potential DNA-binding interactions, but the hKER substitution also had a similar loss of function to the deletion of the KER in vivo. Thus, the context of the KER, including molecular interactions, presence of a viable WHD and structural requirements imposed by the architecture of the CAF-1 complex, is critical for DNA-length sensing and CAF-1 function in vivo.

How might a DNA-length-sensing function of the KER in CAF-1 be relevant during DNA synthesis? One possibility is that extrusion of DNA through PCNA exposes CAF-1 to increasing lengths of naked DNA. The ability of CAF-1 to preferentially bind to a tetrasome length of DNA (*Figure 7a*) could ensure sufficient DNA is available so that as histone binding releases the WHD, there will be sufficient space along the DNA for the subsequent assembly of tetrasomes, while simultaneously protecting newly replicated DNA from spurious binding of other factors (*Ramachandran and Henikoff, 2016*). Loss of the KER can be compensated by the WHD (*Figures 1 and 7b*), but deletion of both the KER and WHD renders yeast with a reduced ability to survive DNA damage or maintain gene silencing (*Figure 7c*). Moreover, the yKER is shorter than the KER in humans and many other eukaryotes, and correspondingly preferentially recognizes shorter DNA (*Figure 6*) than the hKER in the context of yCAF-1, which raises the possibility that the KER serves a spacer function to ensure that tetramers are assembled at specific spacings (*Figure 7d*). Classic SAHs (*Dudola et al., 2017*; *Gáspári et al., 2012*) predominantly use the long rigid helix to bridge two functional domains either as a linker, spacer, or flexible spring (*Wolny et al., 2014*; *Wolny et al., 2017*; *Kwon et al., 2020*; *Bandera et al., 2021*). Therefore, we propose that the KER SAH acts as a DNA-binding physical spacer element and bridge that links with structural precision multiple functional domains within CAF-1 to configure the architecture of CAF-1 for efficient tetrasome assembly after DNA synthesis.

## Materials and methods
### Expression and purification of Cac1 subunits and CAF-1 proteins from insect cells

For expression in insect cells, baculoviruses harboring yCAF-1 subunits and mutants were made using the Gateway technology subcloning system (Thermo Fisher Scientific). The pDONR/Zeo plasmid containing the sequence encoding Cac1 subunit (Cac1, pDONR/Zeo; *Supplementary file 1*) was used for mutagenesis experiments. Briefly, the Cac1, pDONR/Zeo plasmid was linearized via polymerase chain reaction (PCR) with the Q5 DNA polymerase (NEB) and a pair of primers for each Cac1 mutant that anneal to the sequences flanking the section to be modified (*Supplementary file 2*). The primers also had complementary sequences with overhangs: to each other for deletion of Cac1 regions, or to a double-stranded DNA fragment either synthetically manufactured (Integrated DNA

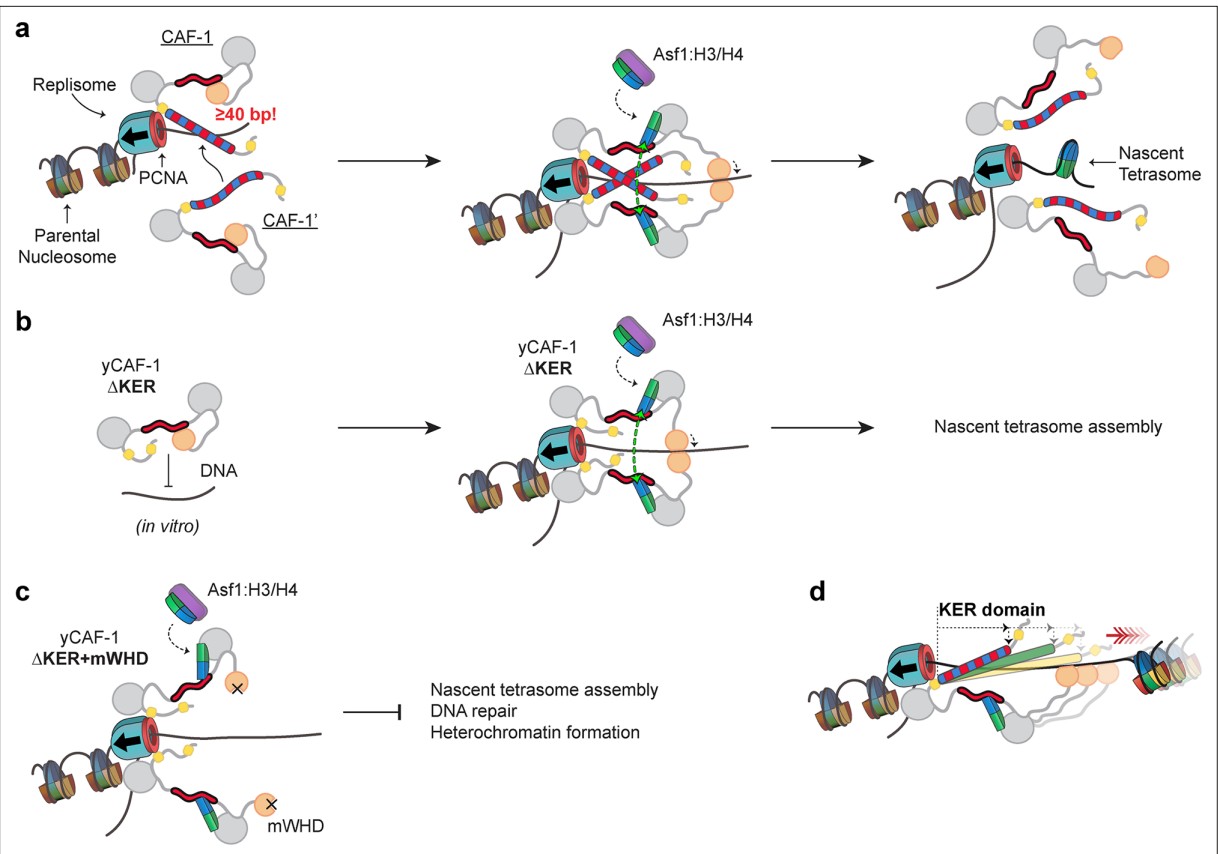

**Figure 7.** Proposed molecular mechanism model of KER-mediated nascent tetrasome assembly by CAF-1. (**a**) The KER safeguards DNA for tetrasome assembly. Because the KER has strong binding affinity toward DNA and is readily competent for binding, recruitment of CAF-1 to DNA through the KER can be an initial transient state prior to assembly of tetrasomes during DNA replication (left panel). Furthermore, the DNA-length selectivity function of the KER equips CAF-1 to bind to free DNA regions that are tetrasome-length (≥40 bp). While CAF-1 is bound to DNA through the KER, CAF-1 can receive newly synthesized H3/H4 dimers from the histone chaperone Anti-silencing Function 1 (Asf1) which in turn facilitates DNA binding of the winged helix domain (WHD; middle panel). The KER and WHD bind cooperatively to DNA which facilitate the recruitment of two copies of the CAF-1–H3/H4 complex to the same DNA vicinity (middle panel). A transient DNA–(CAF-1–H3/H4)$_2$ complex provides the conditions to favor the formation of the H3/H4 tetramer (middle panel, green arrows) following its the deposition on DNA and presumably ejecting CAF-1 from the DNA (right panel). (**b**) Deletion of the KER from yCAF-1 (yCAF-1 ΔKER) impairs binding to DNA in vitro, presumably because the WHD binds more weakly to DNA and is in an autoinhibited state. But in vivo yCAF-1 ΔKER is still competent for tetrasome assembly with minimal sensitivity to DNA damage and defects on heterochromatin formation. (**c**) In contrast, deletion of the KER (ΔKER) in combination with inhibition of the DNA-binding function of the WHD (mWHD) dramatically impairs DNA repair and heterochromatin formation functions of yCAF-1 in vivo. Because in ΔKER+mWHD cells yCAF-1 has no detectable functional DNA-binding domain, tetrasome formation cannot occur efficiently. (**d**) The length of the KER domain in CAF-1 varies across species and it can alter the DNA length recognized by CAF-1 in vitro, which could alter tetrasome assembly during DNA replication.

The online version of this article includes the following figure supplement(s) for figure 7:

**Figure supplement 1.** Cartoon models of KER–DNA association.

Technologies) or PCR amplified from cDNA for the exogenous incorporation of other genes or duplication of Cac1 sequences (*Supplementary file 2*). Circularization of the modified Cac1, pDONR/Zeo (*Supplementary file 1*) plasmid with the desired mutation was done via In-Fusion technology (Takara). Each pDONR/Zeo Cac1 mutant plasmid was verified by Sanger sequencing. Cac1 mutants from the pDONR/Zeo plasmids were then subcloned into the pDEST8 vector via Gateway technology (Thermo Fisher Scientific) followed by Sanger sequencing verification (*Supplementary file 1*). Finally, to generate baculovirus stocks of each Cac1 mutant, we used the Bac-to-Bac system (Thermo Fisher Scientific), where the generated pDEST8 plasmids were transformed into DH10Bac *Escherichia coli* cells (Thermo Fisher Scientific) to generated bacmids competent for baculovirus production in Sf9 cells (Thermo Fisher Scientific) via transfection. Media containing secreted baculovirus from cultured Sf9 cells was stored at 4°C and used for subsequent protein production.

Full-length yCAF-1 and complex mutants were expressed for 72 hr in High Five cells (Thermo Fisher Scientific) infected with a baculovirus stock of Cac1 with a C-terminal Strep II epitope, Cac2 with a C-terminal His$_6$ or Strep II epitope, and Cac3 with a C-terminal 3xFLAG epitope. Purification of the CAF-1 complexes was carried out as before (*Liu et al., 2012*) where cell pellets were homogenized in 20 mM HEPES(N-2-hydroxyethylpiperazine-N'-2-ethanesulfonic acid) pH 7.4, 350 mM NaCl, 1 mM DTT(1,4-dithiothreitol), 10 µg/mL DNase I, 1 mM Na$_3$VO$_4$, 10 mM NaF, 1 mM PMSF(phenylmethyl-sulfonyl fluoride), and a cocktail of protease inhibitors (Tablet EDTA-free, Sigma). Homogenate was clarified by centrifugation at 10,000 × *g* for 45 min at 4°C, followed by affinity chromatography with a StrepTrap HP column (Cytiva) and washed extensively with 20 mM HEPES pH 7.4, 350 mM NaCl, and 0.5 mM TCEP(tris(2-carboxyethyl)phosphine). Protein was eluted with the wash buffer containing 2.5 mM d-Desthiobiotin (MilliporeSigma). Purified yCAF-1 complexes (*Figure 1—figure supplement 1a*) were concentrated with 100,000 MWCO centrifugal concentrators (Sartorius), aliquoted in small volumes, flash frozen in liquid nitrogen, and stored at −80°C.

## Expression and purification of CAF-1 domains and Myosin 7a SAH from *E. coli*

Plasmids for bacterial expression were generated using the Gateway technology subcloning system (Thermo Fisher Scientific). The initial double-stranded DNA insert containing the cDNA that encodes for the protein of interest was obtained either synthetically manufactured (Integrated DNA Technologies) or by PCR amplification from cDNA of a plasmid containing the full-length protein (*Supplementary file 2*). To obtain the double-stranded DNA insert for human KER, *E. coli* codon optimized CHAF1A cDNA was first cloned into a pGEX-6P-1 vector via In Fusion (Takara) (*Supplementary file 3*). For Gateway cloning, the double-stranded DNA inserts were subcloned into the pDONR/Zeo and pDEST566 vectors followed by Sanger sequencing verification of each plasmid (*Supplementary file 1*).

Yeast Cac1 KER constructs (full-length and truncations), human CHAF1A KER, and Myosin 7 SAH cloned into the pDEST566 vector produced an N-terminal His$_6$-MBP-tagged polypeptide with a PreScission protease site downstream of the MBP. Proteins used in CD or EMSA experiments contained an exogenous Tyrosine amino acid as the very last C-terminal residue to facilitate the determination of protein concentration via UV absorption after removal of the His$_6$-MBP tag. Expression of the His$_6$-MBP-tagged proteins were carried out in Rosetta 2 (DE3) pLysS cells cultured in Luria Broth media at 37°C. Bacterial cultures were induced for expression with 0.5 mM IPTG(isopropyl β-d-1-thiogalactopyranoside) when culture reached a 600-nm optical density of 0.8 and let incubate for another 3–4 hr at 37°C. Subsequently, cell pellets were harvested and resuspended in 25 mM Tris pH 7.5, 50 mM NaCl, 10 µg/mL DNaseI, 1 mM PMSF, and a cocktail of protease inhibitors (Tablet EDTA-free, Roche). The lysate was sonicated and then clarified by centrifugation at 10,000 × *g* for 45 min at 4°C. The resulting supernatant was then bound to an agarose Ni-NTA resin (QIAGEN) and incubated for at least 2 hr at 4°C followed by extensive washes with 25 mM Tris pH 7.5, 1 M NaCl, and 1 mM PMSF. Protein was eluted from the Ni-NTA resin with 25 mM Tris pH 7.5, 20 mM NaCl, 500 mM imidazole, and 1 mM PMSF, and further purified by ion exchange chromatography using an SP FF or Source S15 column (Cytiva) with a 30-column volumes salt gradient from 20 mM to 1 M NaCl for elution. Purified His$_6$-MBP-tagged proteins were then concentrated with a 30,000 MWCO centrifugal concentrator (Sartorius) and buffer exchanged into 20 mM HEPES pH 7.5 and 50 mM NaCl for crystallization, or in 50 mM Tris pH 7.5, 150 mM NaCl, 1 mM EDTA(Ethylenediaminetetraacetic acid), 1 mM DTT and 0.01% Triton X-100 for enzymatic removal of the His$_6$-MBP tag with PreScission protease. PreScission protease was added to the His$_6$-MBP-tagged proteins and incubated overnight at 4°C. Cleaved proteins were further purified by ion exchanged as described above, followed by concentration with a 3000 MWCO and buffer exchange into 20 mM HEPES pH 7.5, 150 mM NaCl, and 0.2 mM TCEP (*Figure 1—figure supplement 1b*). Concentrated proteins were aliquoted in small volumes, flash frozen in liquid nitrogen, and stored at −80°C.

The Cac1 WHD construct was expressed and purified as before (*Liu et al., 2016*) using the Cac1 WHD, pGEX-6P-1 plasmid (*Supplementary file 1*).

## Preparation of DNA templates

DNA fragments of varying lengths were designed around the dyad of the 147 bp 601 DNA sequence (*Lowary and Widom, 1998*; *Luger et al., 1997*). Synthetic oligonucleotides for production of dsDNA

(Integrated DNA Technologies) had only one strand containing a Cy5 fluorophore at the 5′ end (**Supplementary file 2**).

Each oligonucleotide was resuspended in 10 mM Tris–HCl and 1 mM EDTA at pH 8.0 (TE Buffer) and non-fluorophore-labeled oligonucleotides were purified using C18 Sep-Pak cartridges (Waters). Annealing of complementary oligonucleotides was done by heating an equimolar combination of each strand at 95°C and then cooling slowly to room temperature in TE Buffer with 5 mM NaCl. Annealed DNA duplexes were purified by ion exchange chromatography using a DEAE column (Tosoh Bioscience) with a gradient of 0–1 M NaCl in TE Buffer. Purified DNA was ethanol precipitated and resuspended in TE Buffer.

The 10–100 bp DNA step ladder was purchased from Promega (Cat. G447A).

## Electrophoretic mobility shift assays

Increasing concentrations of protein that ranged from 9 nM to 10 µM were incubated with 1 or 3 nM of Cy5-labeled DNA in 20 mM HEPES pH 7.4, 150 mM NaCl, and 0.5 mM TCEP for 1 hr on ice. The protein–DNA species were separated by electrophoresis in 0.2× TBE 4%, 5%, 6%, or 10% 59:1 acrylamide:bis-acrylamide native gels for 60 min at 70 V on ice. Fluorescence from the Cy5 fluorophore was detected by imaging the native gels on a c600 (Azure Biosystem) or a Sapphire Biomolecular (Azure Biosystems) imager. To determine dissociation constants between the protein and DNA substrates, the intensity of each DNA band was determined using the AzureSpot software (Azure Biosystems), followed by background subtraction and calculation of the DNA fraction bound. Finally, protein concentrations ([Protein]) and DNA fraction bound values were plotted and the binding curves fitted with **Equation 1** using the Prism software (GraphPad):

$$\text{Fraction bound} = \frac{B_{\max} \times \left[\text{Protein}\right]^h}{K_D^h + \left[\text{Protein}\right]^h} \tag{1}$$

where $h$ corresponds to the Hill coefficient, $K_D$ is the dissociation constant, and $B_{\max}$ represents maximum binding. All EMSA experiments were done at least three times and the reported $K_D$ values and Hill coefficients correspond to the average of the multiple measurements. Error bars represent the standard deviation.

EMSA experiments using the commercial 10 bp DNA step ladder substrate contained 500 nM total DNA and gels were stained with SYBR Green I stain (Invitrogen) prior to imaging.

In the Cy5-DNA ladder assay, the signal of the free DNA of a particular fragment from subsequent protein titration was normalized to the signal of the free DNA in the absence of protein (normalized free DNA). To obtain the $K_{Dapp}$, we generated and fitted binding curves using corresponding fraction bound (**Equation 1**).

## Structure determination and analyses

The His$_6$-MBP-yKER (Cac1 residues 136–225) protein was expressed and purified as described above and protein was concentrated to 25 mg/mL in 20 mM HEPES pH 7.5 and incubated with 200 mM NDSB-256. Protein crystals grew in 0.1 M phosphate/citrate pH 4.2 and 30% PEG 300 using the hanging drop vapor diffusion method at 15°C. Data were collected using a Rigaku Micromax 007 high flux microfocus X-ray generator equipped with a VariMax high flux optic, an AFC11 4-axis goniometer, a Pilatus 200K 2D area detector, and an Oxford cryo-system. Data were initially processed using the HKL-3000R software (HKL Research Inc) and phased by molecular replacement using the structure of MBP (PDB ID: 1PEB) as the search model (**Telmer and Shilton, 2003**). The structure was solved at a resolution of 2.81 Å. The model was built using COOT (**Emsley and Cowtan, 2004**) and refinement was conducted using PHENIX (**Afonine et al., 2012**), to achieve acceptable geometry and stereochemistry. Group TLS refinement was used in the refinement as there were large regions of chain, with much higher than average $B$-factors. Several sections of chain D are poorly defined due to this disorder. The quality of the structure (PDB ID: 8DEI) was analyzed (**Supplementary file 4**) and the RMSD values were calculated using PyMol and COOT. Figures were made using PyMol and Photoshop (Adobe).

## CD spectroscopy

Proteins were prepared for CD at a concentration of 0.1 mg/mL in 10 mM Na-Phosphate Buffer at pH 7.4 and 50 mM NaCl. KER samples containing DNA were mixed at 1:1 molar concentration of

protein and 40 bp 601 DNA. All samples were analyzed in a cuvette with a path length of 1 mm on a Jasco J-815 CD Spectrophotometer equipped with a Lauda Brinkman ecoline RE 106 temperature controller. CD was measured in millidegrees from 185 to 350 nm wavelengths with a bandpass of 1 nm and a step size of 1.0 nm. Six scans of each sample were averaged per experiment with at least three independent replicates. *Equation 2* is used to calculate mean residue ellipticity (MRE) (*Woody and Fasman, 1996*):

$$\text{MRE} = \frac{m^0 \times \frac{\text{MW}}{n-1}}{10 \times L \times C} \tag{2}$$

where $m^0$ is the observed ellipticity in millidegrees, MW is the molecular weight of the protein in g/mol, $n$ is the number of residues of the protein, $L$ is the path length of cell, and $C$ is the concentration of the protein in g/L.

To calculate the fractional helicity of a protein sample, we used the 222 nm wavelength method (*Wei et al., 2014*; *Equation 3*):

$$\text{Fractional helicity} = \frac{\theta_{222}^{\text{exp}} - \theta_{222}^{\text{u}}}{\theta_{222}^{\text{h}} - \theta_{222}^{\text{u}}} \tag{3}$$

where $\theta_{222}^{exp}$ is the experimentally observed MRE at 222 nm of the protein sample, and $\theta_{222}^{u}$ and $\theta_{222}^{h}$ are the MRE at 222 nm of a protein with 0% and 100% helical content which are estimated to be −3000 and −39,000 degxcmxdmol$^{-1}$, respectively.

## Chemical crosslinking

DSS crosslinker (Thermo Pierce) was prepared at 2 mM by dissolution in DMSO. 10 µM yKER (Cac1 residues 136–225 with additional C-terminal Tyrosine) was allowed to incubate with 200 µM DSS or DMSO for 30 min at room temperature in 10 mM Phosphate Buffer at pH 7.5, 150 mM NaCl, and 0.5 mM TCEP. The crosslinking reaction was quenched by addition of 50 mM Tris pH 7.4 and incubated for an additional 15 min. The reactions were resolved in a 4–15% Tris–HCl sodium dodecyl sulfate–polyacrylamide gel electrophoresis (SDS–PAGE; Bio-Rad) and stained with Coomassie Blue.

## Yeast strains and primers

The yeast strains used in this study and their genotypes are fully described in *Supplementary file 3*. Strains used in DNA damage sensitivity assays and western blotting were isogenic to W303-1a (*Thomas and Rothstein, 1989*), while strains used to assay silencing at the *HMR* locus were isogenic to BY4741 (*Brachmann et al., 1998*). Mutations in the *CAC1* gene were made using CRISPR–Cas9 (*Ryan et al., 2016*) to mutate the endogenous *CAC1* (RLF2) locus. Sequences of gRNA and HDR template DNA used to generate each mutant are listed in *Supplementary file 2*. Where indicated, strains were deleted for *RTT106* or *CAC1* using pFA6a-HIS3MX6 (*Longtine et al., 1998*) and pFA6a-KANMX6 (*Bähler et al., 1998*), respectively. Strains for the Okazaki fragment assay were generated by crossing KER domain mutants (RAY165, 264, 221, 245, 233) with the strain YDS12 containing repressible DNA Ligase 1 (*cdc9::tetO7-CDC9 cmv-LacI-NAT*) (*Yeung and Smith, 2020*). After sporulation, haploids were screened using auxotrophic markers, and for presence of KER domain mutations by PCR.

## DNA damage sensitivity assay

Cells were grown in YPD media until reaching mid-log phase (OD 0.8–1.0). They were collected by centrifugation, resuspended in sterile water, and fivefold serially diluted before spotting onto YPD agar plates containing the indicated concentrations of the DNA-damaging drugs Zeocin (Invitrogen R25001) or CPT (Cayman Chemical 11694). Plates were grown for 3 days at 30°C before imaging.

## Measurement of loss of silencing at the *HMR* locus

To observe loss of silencing at the *HMR* locus, cells isogenic to BY4741 were transformed with the plasmid pHMR::P$_{\text{URA3}}$-GFP/URA3 after EcoRI/XhoI digestion (*Laney and Hochstrasser, 2003*). In WT cells, this GFP reporter is silenced, while mutants with loss of silencing express GFP at varying levels that was detected by flow cytometry (*Huang et al., 2005*). 0.5 mL of mid-log phase (OD 0.8–1.0) cells growing in synthetic complete (SC) media containing 2% dextrose were collected by centrifugation, washed twice with ice-cold phosphate buffered saline (PBS), and resuspended in 1 mL of PBS before

analysis on a flow cytometer (BD Biosciences BD LSR II). Cells deleted for *SIR2* (*sir2Δ*) have a severe silencing defect and were used as a positive control. As indicated in *Figure 1—figure supplement 2f*, a gate containing <1% of WT cells and >97% of *sir2Δ* cells was drawn and used to identify the percent of cells with loss of silencing. As previously observed, the percentage of WT cells with loss of silencing varied from ~0.3% to 2.5% across experiments. Data are presented as the average ± standard deviation of at least three experiments performed on independent yeast colonies.

## Okazaki fragment length assay

Okazaki fragment labeling reactions were carried out as previously described (*Smith and Whitehouse, 2012*). Briefly, strains carrying doxycycline-repressible CDC9 were treated with doxycycline for 2.5 hr. Following preparation of genomic DNA, 3′ ends were labeled by extension with Klenow exo- (NEB) and [alpha]32P-dCTP, separated in 1.3% alkaline agarose gels, and transferred to a Hybond-N membrane for visualization.

## Western blot for Cac1-FLAG

1 OD of cells with an endogenous C-terminal FLAG tag on *CAC1* were grown in YPD to mid-log phase, collected, flash frozen in liquid nitrogen, resuspended in 100 µL of modified Laemmli buffer (*Horvath and Riezman, 1994*), and boiled for 5 min. Proteins were separated on a 10% SDS–PAGE gel and western blotting was performed using anti-FLAG M2 (Sigma F3165) and anti-GAPDH (Sigma A9521).

# Acknowledgements

We appreciate the contributions of Dr. Ying-Chih Chi for preparing the initial CHAF1A expression plasmid and Dr. Mark Hochstrasser for providing the pHMR::PURA3-GFP/URA3 plasmid to us. We thank the Biophysical core facility at CU-Anschutz for experimental advice and assistance, as well as the Structural Biology Shared Resources of the University of Colorado Cancer Center supported by NIH P30 CA046934. This work was supported by NIH R01 GM135604 and NIH Shared instrumentation grant S10 OD012033 to MEAC., R35 GM134918 to DJS., R01 CA95641 and R35 GM139816 to JKT. RRA was supported by a Medical Scientist Training Program grant from the National Institute of General Medical Sciences of the National Institutes of Health under award number: T32 GM007739 to the Weill Cornell/Rockefeller/Sloan Kettering Tri-Institutional MD-PhD Program, and a diversity supplement on grant RO1 GM64475 to JKT.

# Additional information

### Competing interests

Ruben Rosas: JKT, eLife Senior Editor. The other author declares that no competing interests exist.

### Funding

| Funder | Grant reference number | Author |
|--------|------------------------|--------|
| National Institutes of Health | R01GM135604 | Mair EA Churchill |
| National Institutes of Health | R35 GM134918 | Duncan J Smith |
| National Institutes of Health | R01 CA95641 | Jessica K Tyler |
| National Institutes of Health | R35 GM139816 | Jessica K Tyler |
| National Institutes of Health | T32 GM007739 | Rhiannon R Aguilar |
| National Institutes of Health | R01 GM64475 | Jessica K Tyler |

| Funder | Grant reference number | Author |
| --- | --- | --- |
| National Institute of General Medical Sciences | S10 OD012033 | Mair EA Churchill |
| National Cancer Institute | NIH P30 CA046934 | Mair EA Churchill |

The funders had no role in study design, data collection, and interpretation, or the decision to submit the work for publication.

## Author contributions
Ruben Rosas, Data curation, Formal analysis, Validation, Investigation, Visualization, Methodology, Writing – original draft, Writing – review and editing; Rhiannon R Aguilar, Data curation, Formal analysis, Investigation, Visualization, Methodology, Writing – original draft, Writing – review and editing; Nina Arslanovic, Data curation, Investigation, Methodology, Writing – review and editing; Anna Seck, Data curation, Investigation; Duncan J Smith, Conceptualization, Data curation, Supervision, Investigation; Jessica K Tyler, Conceptualization, Supervision, Funding acquisition, Project administration, Writing – review and editing; Mair EA Churchill, Conceptualization, Data curation, Formal analysis, Supervision, Funding acquisition, Validation, Visualization, Methodology, Writing – original draft, Project administration, Writing – review and editing

## Author ORCIDs
Rhiannon R Aguilar ⓘ http://orcid.org/0000-0001-5723-608X
Nina Arslanovic ⓘ http://orcid.org/0000-0001-5527-0874
Duncan J Smith ⓘ http://orcid.org/0000-0002-0898-8629
Jessica K Tyler ⓘ http://orcid.org/0000-0001-9765-1659
Mair EA Churchill ⓘ http://orcid.org/0000-0003-0862-235X

## Decision letter and Author response
Decision letter https://doi.org/10.7554/eLife.83538.sa1
Author response https://doi.org/10.7554/eLife.83538.sa2

# Additional files

## Supplementary files
- Supplementary file 1. List of plasmids.
- Supplementary file 2. List of synthetic DNA oligonucleotides and primers.
- Supplementary file 3. List of yeast strains.
- Supplementary file 4. Statistics for crystallographic data collection and refinement.
- MDAR checklist

## Data availability
Diffraction data have been deposited in PDB under the accession code 8DEI. All data generated or analyzed during this study are included in the manuscript, and Source Data Zip archives: Source_Data_Fig1b-i.zip, Source_Data_Fig1j-o_Supp.zip, Source_Data_Figs2-3.zip, Source_Data_Figs4-6.zip.

The following dataset was generated:

| Author(s) | Year | Dataset title | Dataset URL | Database and Identifier |
| --- | --- | --- | --- | --- |
| Churchill MEA, Rosas R | 2023 | Diffraction data | https://www.rcsb.org/structure/8DEI | RCSB Protein Data Bank, 8DEI |

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
