## [Editor Report]

The work is an important contribution that advances understanding of CAF-1 function by identifying the KER region as that responsible for DNA size-selective binding of the complex. X-ray crystallography shows the KER region forms an unusual single α-helix (SAH) structure that binds DNA on its own in a size-selective manner. Mutations in the KER helix compromise chromatin assembly and give DNA damage sensitive phenotypes.

---

## [Decision Letter]

**Decision letter after peer review:**

Thank you for submitting your article "A novel Single Α-Helix-DNA-binding domain in CAF-1 promotes gene silencing and DNA damage survival through tetrasome-length DNA selectivity and spacer function" for consideration by *eLife*. Your article has been reviewed by 3 peer reviewers, and the evaluation has been overseen by a Reviewing Editor and Kevin Struhl as the Senior Editor. The following individual involved in review of your submission has agreed to reveal their identity: Zhiguo Zhang (Reviewer #1).

Essential revisions:

The work advances the understanding of CAF-1 function by identifying the KER region as that responsible for DNA size-selective binding of the complex. X-ray crystallography shows the KER region forms an unusual single α-helix (SAH) structure that binds DNA on its own in a size-selective manner. Mutations in the KER helix compromise chromatin assembly and give DNA damage sensitive phenotypes.

1) The drug resistance and gene silencing phenotypes are inherently indirect measures of the most important claim of this work, that KER is a molecular ruler for DNA for the purpose of ensuring sufficiently large templates deposition of histone H3/H4 cargoes. Therefore, this study would be of greater impact if the authors more directly tested this measurement idea in assays that directly assess histone deposition. There are multiple options. Since you have in hand recombinant wild-type and mutant CAF-1 complexes, one could examine the number and/or spacing of nucleosomes formed during in vitro deposition reactions. Complementary in vivo experiments using your existing mutant strains could be based on the finding that CAF-1 is particularly important for histone deposition onto nascent Okazaki fragments during DNA replication (Smith and Whitehouse, 2012; pmid: 22419157), and that the spacing pattern of nucleosomes on this DNA is greatly perturbed in cac1-delete cells.

2) You may consider performing a molecular docking experiment to provide insight as to how the KER SAH associates with DNA. How does it bind one DNA helix? Do they associate in a linear fashion? How do two DNA strands bind as suggested by the EMSA gels?

*Reviewer #1 (Recommendations for the authors):*

Rosas et al. performed the structural and functional studies of KER region of yeast Cac1, the large subunit of chromatin assembly factor 1 (CAF-1). They found that KER region forms a single α helix motif (SAH). Unlike SAH found in other proteins, they showed that SAH motif of both yCac1 and its human counterpart, p150, binds DNA in sequence length dependent manner. They found that depletion of KER in yeast cells when combined with WHD region of Cac1, another region that binds DNA, completely destroyed the function of Cac1 in heterochromatin silencing and in response to DNA damage agents. Furthermore, the KER region and WHD region bind DNA with different affinity and likely have distinct function in cells. Then, they performed a series of mutagenesis studies on yeast KER and identified a couple of key amino acids important for the function. Overall, almost all experiments are well designed and controlled, and the results largely support the conclusion. I have only one major concern and several minor concerns described below.

The results that hKER cannot complement the function of yeast KER are very puzzling considering the following results shown (1) both yKER and hKER bind DNA; (2) Cac1 mutant with 2KER behaves like wild type Cac1. These results suggest that in addition to bind DNA, the KER region may have additional function. Other possibilities include, (1) the Cac1 mutant contain hKER was not expressed, (2) the mutant CAF-1 with hKER cannot bind DNA. The authors need to test some of these possibilities to strengthen conclusion.

*Reviewer #2 (Recommendations for the authors):*

Specific comments:

Figure 1.

– The data shown in this figure would be easier for readers to compare if gels from experiments with the same titrant proteins were stacked vertically above one another.

– The exact protein concentrations used in each set of experiments (Figure 1b-I and Figure 1 Supp. 2a-b and Figure 3) is not stated in the figure legend (preferable) or in the Materials and methods.

– Also, the authors should briefly comment on why a different set of DNA sizes was used for the yCAF-1 and y-KER experiments.

– The indication of wild type (WT) and Cac1 WT in Figure 1n and o, I found a bit confusing, perhaps use "Cac1 WT" for both.

The authors should provide a detailed linear map for Cac1 delineating the position of KER, and the precise location of the mutants and deletions used in the work.

Figure 1 – supplementary 1 – the authors should indicate the proteins shown are derived from isolated yCAF-1 complexes from insect cells. Also adding arrows indicating Cac2 and Cad3 band would help readers note the two closely positioned bands.

Line 80-82: The presentation can be improved here, the wording "we first expressed and purified recombinant tri-subunit yCAF-1, KER (Cac1 residues 136-225, yKER), and WHD (Cac1 residues 457-606, yWHD)" is confusing. Perhaps "…tri-subunit yCAF-1 complex and the peptides KER…"

Line 440 Given the prior discussion, the phrase " … and / or orientation of the KER helix in CAF-1…" should be included here.

The authors may consider performing a molecular docking experiment to provide insight as to how the KER SAH associates with DNA. How does it bind one DNA helix? Do they associate in a linear fashion? How do two DNA strands bind as suggested by the EMSA gels?

*Reviewer #3 (Recommendations for the authors):*

The δ-KER version of yCAF-1 appears poorly behaved in the gel shift assay in Figure 1m, perhaps, as suggested by the authors, due to aggregation. Fortunately, the subsequent analyses of smaller deletions largely resolves this problem: the δ-middle-A mutant protein appears to not bind DNA efficiently, without appearing to cause aggregates at the bottom of the gel well (Figure 3h). Therefore, I suggest Figure 1m be moved to the supplemental data, and perhaps a wider titration of the δ-middle-A protein could take its place.

Figure 4: Some of the Cy5 DNA ladder profiles appear very similar to those observed with wild-type yCAF-1. Specifically, by what statistical measures are the ED::GSL (Figure 4e) and KER::hKER (Figure 6c) mutants distinct from wt CAF-1? Also, Since some of the differences among these curves are subtle, it would greatly help the reader to have the curves for wild-type CAF-1 alongside all mutants examined, e.g. in Figure 5. Or, combine all these titrations into a single figure?

To test the contribution of the KER domain, it would have been instructive to analyze δ-middle-A CAF-1 complex using these same titrations in the Cy5 DNA ladder assay. Also, some mutants are not fully characterized: the helix altering delta145-9 and delta225-6 mutants (Figure 5d-e) should be tested in the biochemical assays, and the +N-half mutant needs to be tested in biological assays.

The 2xKER and +N-half mutant CAF-1 complexes bind short DNA fragments (20-30 bp) better than wt, consistent with the observed bypass by 2xKER of the deltaWHD phenotypes in the biological assays. What is the proposed molecular mechanism, that is, why do the authors think increasing the length of KER would provide affinity for a broader range of DNA sizes? And why would the longer human hKER not be able to do this?

Regarding the analysis in Figure 4f- why not show the data for the yWHD domain?

---

## [Author Response]

Essential revisions:The work advances the understanding of CAF-1 function by identifying the KER region as that responsible for DNA size-selective binding of the complex. X-ray crystallography shows the KER region forms an unusual single α-helix (SAH) structure that binds DNA on its own in a size-selective manner. Mutations in the KER helix compromise chromatin assembly and give DNA damage sensitive phenotypes.1) The drug resistance and gene silencing phenotypes are inherently indirect measures of the most important claim of this work, that KER is a molecular ruler for DNA for the purpose of ensuring sufficiently large templates deposition of histone H3/H4 cargoes. Therefore, this study would be of greater impact if the authors more directly tested this measurement idea in assays that directly assess histone deposition. There are multiple options. Since you have in hand recombinant wild-type and mutant CAF-1 complexes, one could examine the number and/or spacing of nucleosomes formed during in vitro deposition reactions. Complementary in vivo experiments using your existing mutant strains could be based on the finding that CAF-1 is particularly important for histone deposition onto nascent Okazaki fragments during DNA replication (Smith and Whitehouse, 2012; pmid: 22419157), and that the spacing pattern of nucleosomes on this DNA is greatly perturbed in cac1-delete cells.

Thank you for the suggestion of approaches to obtain data that more directly addresses changes in nucleosome assembly due to CAF-1 KER mutants. We considered using an in vitro nucleosome assembly assay, such as the reconstitution of nucleosomes onto gapped DNA using purified components developed by Kadyrova et al., 2013 (doi: 10.4161/cc.26310). However, they found defects only in the amount of nucleosome assembly and not changes in nucleosome spacing without CAF-1. In addition, we didn’t have the system set up and knew that it would be unlikely to produce data in the time needed for a revision of the manuscript, or even show spacing changes in nucleosomes at all. Therefore, we chose an assay system in yeast that already has been used to assess the impact of CAF-1 DNA binding mutants on nucleosome assembly (Smith and Whitehouse, 2012; pmid: 22419157 and Mattiroli *et al.*, 2017 doi: 10.7554/*eLife*.22799). This approach, developed by Smith and Whitehouse, uses a degradable Ligase I system in yeast, which reveals Okazaki fragment lengths, and shows a defect when CAF-1 activity is knocked out (Smith and Whitehouse, 2012). This assay also showed that mutations or deletions in the Cac1 WHD DNA binding domain, led to increased lengths of Okazaki fragments (Mattiroli *et al.*, 2017). As the WHD DBD impacts Okazaki fragment lengths, we reasoned that mutations in the KER DBD might also.

We generated numerous new yeast strains that included the degradable Ligase I system and collaborated with Dr. Duncan Smith of (Smith and Whitehouse, 2012; pmid: 22419157) to detect nascent Okazaki fragments in various CAC1 mutants in strains that were *RTT106* or *rtt106∆*. We found that the Okazaki fragment lengths from *cac1∆* yeast were larger and less discrete than from *CAC1* yeast (as Dr Smith published previously) and that the Okazaki fragments from the *cac1∆ rtt106∆* strain were barely detectable, presumably because they were too long to be resolved on the gel. However, the assay did not have sufficient resolution to detect changes between the Okazaki fragment length distribution between wild type CAC1 or the ∆KER, ∆middle-A and 2xKER mutants of CAC1, in either the *RTT106* or *rtt106∆* background. Therefore, we were unable to detect direct effects of the KER mutants on Okazaki-fragment lengths. We considered using the combination of KER mutants with the WHD mutants, but as this would not directly assess the effects of the KER mutants and CAF-1 proteins lacking the KER and the WHD don’t bind to DNA (Figure 3 in Mattiroli et al., 2017), we didn’t pursue it. As the complete deletion of the KER, shortening of the KER and lengthening of the KER did not give detectable changes in this assay, we also did not pursue the other mutants tested in the manuscript. Although, we are disappointed the experiment did not reveal effects that we had hoped for, this experiment provides support for the redundant functions of CAF-1 and Rtt106 in nucleosome assembly, which has not been shown using this assay. As such, we have added Figure 1—figure supplement 1g and text to the Results section, methods section and strain table. We have included Prof. Duncan Smith and his student Anne Seck as authors.

Added text lines 195 to 207: “Finally, to assess the impact of deleting the KER more directly on nucleosome assembly in vivo, we examined histone deposition onto nascent Okazaki fragments during DNA replication as we have shown previously that the length of Okazaki fragment lengths are determined by histone deposition into nucleosomes and is disrupted upon deletion of *CAC1* (Smith and Whitehouse, 2012). We compared CAF-1 mutants in the WT yeast background and in yeast lacking Rtt106. We found that the Okazaki fragment length distributions of the ∆KER mutant was indistinguishable from that of WT while that of *cac1∆* was disrupted (Figure 1—figure supplement Figure 1—figure supplement 3g). That we did not detect effects on Okazaki-fragment lengths for the yCAF-1 mutants lacking the intact KER is consistent with the results of the viability and silencing assays for KER mutants, which also retained the WHD. Strikingly, the Okazaki fragments from *rtt106∆ cac1∆* yeast were highly disrupted (Figure 1—figure supplement Figure 1—figure supplement 3g) further highlighting the redundancy between Rtt106 and Cac1 for assembling histones onto newly replicated DNA. Therefore, t”

2) You may consider performing a molecular docking experiment to provide insight as to how the KER SAH associates with DNA. How does it bind one DNA helix? Do they associate in a linear fashion? How do two DNA strands bind as suggested by the EMSA gels?

We appreciate the interest in understanding the mode of DNA recognition of the KER and suggestions to do docking studies. Indeed, we have invested a large amount of time and effort to obtain the structure of the KER-DNA complex and we had performed molecular docking. However, we opted previously not to present any detailed models in this manuscript, which may give the reader the sense that we favor a particular mode of binding, when there are several viable alternatives. We have now added Figure 7—figure supplement 1, which shows two possible modes of KER-DNA binding. For example, the KER binds independently and cooperatively to DNA in a final ratio related to the DNA length, suggesting a model by which the KER “crosses the DNA”, much like a leucine-zipper DBD. Alternatively, the positively-charged face of the KER could align with the DNA in the context of intact CAF-1. However, for the KER in the context of the intact CAF-1 complex, the number of CAF-1-DNA complexes observed is 2 and this number does not change with alterations to DNA length, which suggests that DNA binding is constrained by other interactions within the complex.

Edited text within 596 to 612: “the KER SAH is capable of binding to DNA independently and cooperatively in a final ratio related to the DNA length with an estimated site size as short as 20 bp, which is typical of many DBDs. Many α helical DNA binding motifs, including leucine zippers and helix-loop-helix motifs, bind across the DNA within the major groove (Luscombe et al., 2000; Wolberger, 2021; Churchill and Travers, 1991). Alternatively, the positively-charged face of the KER could align with the DNA, similar to the long helices that lie parallel to the DNA exist in chromatin remodelers, such as the HSA and post-HSA domains in the actin related proteins (Arp4 and Arp8) of INO80 (Knoll et al., 2018; Baker et al., 2021). However, these helices simultaneously interact extensively with other polypeptides in addition to the DNA (Knoll et al., 2018; Baker et al., 2021). In the context of the intact CAF-1 complex, the number of CAF-1-DNA complexes observed is limited to 2 and this number does not change with different DNA lengths, which suggests that KER-DNA binding is constrained by other interactions within the complex. Whether the KER binds along the length of the DNA or engages only a short stretch of the DNA in a similar manner to the other helical motifs is not clear. Our results are consistent with aspects of these models (Figure 7—figure supplement 1a and b), as the middle region of the KER confers the ability to bind to DNA, and the positively charged amino acids along one face of the SAH DBD would be suitable for electrostatic steering and recognition of longer segments of DNA.”

Reviewer #1 (Recommendations for the authors):Rosas et al. performed the structural and functional studies of KER region of yeast Cac1, the large subunit of chromatin assembly factor 1 (CAF-1). They found that KER region forms a single α helix motif (SAH). Unlike SAH found in other proteins, they showed that SAH motif of both yCac1 and its human counterpart, p150, binds DNA in sequence length dependent manner. They found that depletion of KER in yeast cells when combined with WHD region of Cac1, another region that binds DNA, completely destroyed the function of Cac1 in heterochromatin silencing and in response to DNA damage agents. Furthermore, the KER region and WHD region bind DNA with different affinity and likely have distinct function in cells. Then, they performed a series of mutagenesis studies on yeast KER and identified a couple of key amino acids important for the function. Overall, almost all experiments are well designed and controlled, and the results largely support the conclusion. I have only one major concern and several minor concerns described below.The results that hKER cannot complement the function of yeast KER are very puzzling considering the following results shown (1) both yKER and hKER bind DNA; (2) Cac1 mutant with 2KER behaves like wild type Cac1. These results suggest that in addition to bind DNA, the KER region may have additional function. Other possibilities include, (1) the Cac1 mutant contain hKER was not expressed, (2) the mutant CAF-1 with hKER cannot bind DNA. The authors need to test some of these possibilities to strengthen conclusion.

Regarding the possibilities suggested:

1) We had included western blot analyses (Figure 1—figure supplement 1) that all of the CAF-1 mutants are expressed in yeast to similar extents.

2) We had shown that hKER in the context of intact CAF-1 (yCAF1 KER::hKER ) binds to DNA (Figure 6c) with a similar affinity as yCAF-1 (Figure 4 a and f).

Therefore, we believe that neither of these possibilities is the reason for the interesting differences that we observed in vivo for the yCAF1 KER::hKER mutant. In fact, like yCAF1 KER::hKER, small deletions within the KER also give rise to similar phenotypes as the ∆KER. These findings led us to propose that the SAH structure of the KER functions as a “… physical spacer element and bridge that links with structural precision multiple functional domains within CAF-1 to configure the architecture of CAF-1 for efficient tetrasome assembly after DNA synthesis” (pg 19). Further details on the molecular mechanism of this proposed “bridge” function of the KER SAH are beyond the scope of this article and may require a more detailed understanding of the overall structure of the entire CAF-1 complex.

Reviewer #2 (Recommendations for the authors):Specific comments:Figure 1.– The data shown in this figure would be easier for readers to compare if gels from experiments with the same titrant proteins were stacked vertically above one another.– The exact protein concentrations used in each set of experiments (Figure 1b-I and Figure 1 Supp. 2a-b and Figure 3) is not stated in the figure legend (preferable) or in the Materials and methods.– Also, the authors should briefly comment on why a different set of DNA sizes was used for the yCAF-1 and y-KER experiments.

Thank you for the suggestions. We have now arranged the EMSA images vertically in the edited Figure 1. In addition, we have put the concentrations of the protein below each lane of every EMSA image (edited Figures 1- 3 and relevant figure supplements) so that it will be easier for the reader to compare concentrations. To explain why we used different DNA lengths for yCAF-1 and KER experiments, we found in preliminary testing, that no DNA binding was observed for yCAF-1 to the shortest fragments 10 bp and 20 bp, and therefore we didn’t use them in further analyses. Otherwise the same fragments were tested.

– The indication of wild type (WT) and Cac1 WT in Figure 1n and o, I found a bit confusing, perhaps use "Cac1 WT" for both.

The statistical comparisons are being made to the Cac1 WT strain in the rtt106∆ background. To make a clearer distinction between the strain backgrounds, we have shaded the rtt106∆ strains in Figure 1o, and all of the other panels where the two strains are shown adjacent to each other.

The authors should provide a detailed linear map for Cac1 delineating the position of KER, and the precise location of the mutants and deletions used in the work.

Thank you for the suggestion. Figure 1—figure supplement 1 has been added to contain detailed linear maps for all the Cac1 constructs used in this work.

Figure 1 – supplementary 1 – the authors should indicate the proteins shown are derived from isolated yCAF-1 complexes from insect cells. Also adding arrows indicating Cac2 and Cad3 band would help readers note the two closely positioned bands.

We appreciate this suggestion of clarification. We have changed the Figure legend for Figure 1 supplement 1 at lines 125 and 126 to “Coomassie Blue of the indicated protein domains purified from bacteria (b and c) or yCAF-1 complexes purified from insect cells”. We have also added the arrows.

Line 80-82: The presentation can be improved here, the wording "we first expressed and purified recombinant tri-subunit yCAF-1, KER (Cac1 residues 136-225, yKER), and WHD (Cac1 residues 457-606, yWHD)" is confusing. Perhaps "…tri-subunit yCAF-1 complex and the peptides KER…"

Thank you for the suggestion. We used “isolated domains” instead of peptide, because these were produced recombinantly and were not synthetically. We have changed the manuscript at line 83-84 to “… tri-subunit yCAF-1 complex, and the isolated domains KER….”

Line 440 Given the prior discussion, the phrase " … and / or orientation of the KER helix in CAF-1…" should be included here.

Thank you for suggesting this. We have changed the manuscript at line 458 to: “We conclude that a very specific length and/or orientation of the yKER helix in CAF-1 is critical to overcome DNA damage and maintain gene silencing in vivo.”

The authors may consider performing a molecular docking experiment to provide insight as to how the KER SAH associates with DNA. How does it bind one DNA helix? Do they associate in a linear fashion? How do two DNA strands bind as suggested by the EMSA gels?

We appreciate the interest in understanding the mode of DNA recognition of the KER and suggestions to do docking studies. Indeed, we have invested a large amount of time and effort to obtain the structure of the KER-DNA complex and we have done some docking. However, we opted not to present any detailed models in this manuscript, which may give the reader the sense that we favor a particular mode of binding, when there are several viable alternatives. Instead, we have added Figure 7—figure supplement 1, which shows two possible modes of KER-DNA binding. For example, the KER binds independently and cooperatively to DNA in a final ratio related to the DNA length, suggesting a model by which the KER “crosses the DNA”, much like a leucine-zipper DBD. Alternatively, the positively-charged face of the KER could align with the DNA in the context of intact CAF-1. However, for the KER in the context of the intact CAF-1 complex, the number of CAF-1-DNA complexes observed is 2 for the 40 bp DNA length, which suggests that DNA binding is constrained by other interactions within the complex.

Edited text within 593 to 612 “The molecular mechanism of KER-DNA recognition requires both key positively charged residues and α helical conformation in the Cac1 middle-A section (Figure 3). Although CAF-1 prefers to bind to tetrasome-length DNA (Luger et al., 1997; Donham, Scorgie, and Churchill, 2011), the KER SAH is capable of binding to DNA independently and cooperatively in a final ratio related to the DNA length with an estimated site size as short as 20 bp, which is typical of many DBDs. Many α helical DNA binding motifs, including leucine zippers and helix-loop-helix motifs, bind across the DNA within the major groove (Luscombe et al., 2000; Wolberger, 2021; Churchill and Travers, 1991). Alternatively, the positively-charged face of the KER could align with the DNA, similar to the long helices that lie parallel to the DNA exist in chromatin remodelers, such as the HSA and post-HSA domains in the actin related proteins (Arp4 and Arp8) of INO80 (Knoll et al., 2018; Baker et al., 2021). However, these helices simultaneously interact extensively with other polypeptides in addition to the DNA (Knoll et al., 2018; Baker et al., 2021). In the context of the intact CAF-1 complex, there are fewer CAF-1-DNA complexes observed, which suggests that KER-DNA binding is constrained by other interactions within the complex. Whether the KER binds along the length of the DNA or engages only short stretches of the DNA in a similar manner to the other helical motifs is not clear. Our results are consistent with aspects of both of these models (Figure 7—figure supplement 1a and b), as the middle region of the KER confers the ability to bind to DNA, and the positively charged amino acids along one face of the SAH DBD would be suitable for electrostatic steering as well as recognition of long segments of DNA.”

Reviewer #3 (Recommendations for the authors):The δ-KER version of yCAF-1 appears poorly behaved in the gel shift assay in Figure 1m, perhaps, as suggested by the authors, due to aggregation. Fortunately, the subsequent analyses of smaller deletions largely resolves this problem: the δ-middle-A mutant protein appears to not bind DNA efficiently, without appearing to cause aggregates at the bottom of the gel well (Figure 3h). Therefore, I suggest Figure 1m be moved to the supplemental data, and perhaps a wider titration of the δ-middle-A protein could take its place.

Thank you for the suggestion. However, because the ∆KER mutation is referred throughout the paper in the in vivo assays, we consider that it should remain in the main figure as it still provides evidence that the KER is necessary for the function and/or stability of the yCAF-1 complex. Moreover, the rationale for producing the ∆middle-A mutation came from the results that are presented in Figure 3. Therefore, we feel it would be better to leave the figure placements as they are.

Figure 4: Some of the Cy5 DNA ladder profiles appear very similar to those observed with wild-type yCAF-1. Specifically, by what statistical measures are the ED::GSL (Figure 4e) and KER::hKER (Figure 6c) mutants distinct from wt CAF-1? Also, Since some of the differences among these curves are subtle, it would greatly help the reader to have the curves for wild-type CAF-1 alongside all mutants examined, e.g. in Figure 5. Or, combine all these titrations into a single figure?

We agree, but there are some differences that we observed and wished to compare for the mutants. It was be difficult to put all of the results in one figure due to the rationale for the design of the mutants and the flow of the manuscript. Therefore, have left the EMSA and analyses where they were. To address the comparison suggested by this reviewer, we added a comparison graph. We considered many approaches for quantitating these differences to make these comparisons easier for the reader. We settled on measuring the concentration at 50% bound for each of the fragments and summarized the results in Figure 4f and g. The new Figure 4g shows the rate (slope) by which the apparent dissociation constant changes from 40 bp to 50 bp, which is one way to visualize a difference in the threshold of maximum binding affinity for the 30-50 bp fragments. When compared to yCAF-1, the ED::GSL and KER::hKER have a greater preference for the longer 50 bp DNA compared to 40 bp. To demonstrate that this threshold is significantly different for these two mutants, we have looked at the rate (slope) by which the apparent dissociation constant changes from 40 bp to 50 bp. Using one-way ANOVA, the rate change from 40 bp to 50 bp is significantly different for only ED::GSL and KER::hKER when compared to yCAF-1 (now Figure 4g). This has been introduced and described in the manuscript at line 381-386. “We observed a threshold effect whereby CAF-1 binds to DNA of 40 bp in length length or longer with a similar KDapp, but there were mutants that had increased binding affinity for 50 bp compared to 40 bp. This can be seen by plotting the rate of the apparent dissociation constant change from 40 bp to 50 bp, which allows for these thresholds to be easily distinguished (Figure 4f and g).”

To test the contribution of the KER domain, it would have been instructive to analyze δ-middle-A CAF-1 complex using these same titrations in the Cy5 DNA ladder assay. Also, some mutants are not fully characterized: the helix altering delta145-9 and delta225-6 mutants (Figure 5d-e) should be tested in the biochemical assays, and the +N-half mutant needs to be tested in biological assays.

The CAF-1 Δmiddle A doesn’t bind to DNA at all at 250 nM concentrations (Figure 3h), which are in the range of the Cy5 ladder assay. Therefore, we did not test any lower concentrations. To make this point clearer, we have included a version of Figure 3h using 760 nM of protein concentration (Figure 3—figure supplement 1) where we observed no DNA shifts for ∆middle-A. Reference to this new figure has been introduced in the manuscript. With regard to testing the ∆145-149 and ∆225-226 mutants, we had already found that larger fragments, N-half and C-half, were unable to bind to DNA, and the middle-A could bind to DNA without those regions. Therefore, recombinant yCAF-1 complexes with ∆145-149 or ∆225-226 deletions were not considered for insect cell production and DNA binding testing. Regarding testing the +N-half in yeast, we had not done this because the 2XKER tested for the effect of very long KER and the hKER represents a KER that is slightly (20 aa) longer KER and believed that these together represent two extremes and that the +N-half was intermediate to those.

The 2xKER and +N-half mutant CAF-1 complexes bind short DNA fragments (20-30 bp) better than wt, consistent with the observed bypass by 2xKER of the deltaWHD phenotypes in the biological assays. What is the proposed molecular mechanism, that is, why do the authors think increasing the length of KER would provide affinity for a broader range of DNA sizes? And why would the longer human hKER not be able to do this?

In isolation and unlike the yCAF-1 complex, the KER binds well to 20 bp and 30 bp fragments of DNA (Table 1, Figure 4b and g). This suggests that something in the CAF-1 complex is limiting the DNA binding to short fragments, possibly steric hindrance due to the presence of the rest of the CAF-1 complex. In the case of the yCAF-1 2xKER and +N-half mutants, the overall KER region is substantially larger than in the intact yCAF-1, 37 and 90 residues longer respectively. These longer KER domains could reduce the physical constrains imposed in the CAF-1 complex around the KER-DNA-binding region, therefore increasing the accessibility and binding to short DNA fragments. In the case of the hKER, which is only 20 residues longer than the yKER this increase in the length of the KER domain in the yCAF-1 yKER::hKER mutant is not sufficient to observe an increased binding affinity to short fragments of DNA or compensate for the loss of the WHD.

Regarding the analysis in Figure 4f- why not show the data for the yWHD domain?

The depletion of the Cy5-DNA ladder fragments for the yWHD (Figure 4c) shows that the DNA binding of the WHD is not substantially impacted by the length of DNA, i.e. the WHD does not exhibit DNA length selectivity. Therefore, the yWHD was not considered for the analysis on the estimates of DNA-length-dependent dissociation constants.